# ATOMWORLD: A BENCHMARK FOR EVALUATING SPATIAL REASONING IN LARGE LANGUAGE MODELS ON CRYSTALLINE MATERIALS

## ABSTRACT

Large Language Models (LLMs) excel at textual reasoning and are beginning to develop spatial understanding, prompting the question of whether these abilities can be combined for complex, domain-specific tasks. This question is essential in fields like materials science, where deep understanding of 3D atomic structures is fundamental. While initial studies have successfully applied LLMs to tasks involving pure crystal generation or coordinate understandings, a standardized benchmark to systematically evaluate their core reasoning abilities across diverse atomic structures has been notably absent. To address this gap, we introduce the AtomWorld benchmark to evaluate LLMs on tasks based in Crystallographic Information Files (CIFs), a standard structure representation format. These tasks, including structural editing, CIF perception, and property-guided modeling, reveal a critical limitation: current models, despite establishing promising baselines, consistently fail in structural understanding and spatial reasoning. Our experiments show that these models make frequent errors on structure modification tasks, and even in the basic CIF format understandings, potentially leading to cumulative errors in subsequent analysis and materials insights. By defining these standardized tasks, AtomWorld lays the ground for advancing LLMs toward robust atomic-scale modeling, crucial for accelerating materials research and automating scientific workflows.

## 1 INTRODUCTION

A Crystallographic Information File (CIF) (Hall et al., 1991) is the standard format for storing crystallographic structural data. At the most basic level, a CIF can model the ideal, periodic arrangement of atoms in a bulk material. For more realistic scenarios, CIFs can also model defects, molecules in defect sites, stacked heterostructures, etc. Suppose that there are three stages for an LLM to reason with CIF files: **motor skills**, **perceptual skills** and **cognitive skills**. **Motor skills** are about the mechanics of geometry - being able to add, move, rotate, or insert atoms consistently within a structure. **Perceptual skills** are about recognising patterns - seeing motifs, detecting symmetry or connectivity, and being able to relate this structure to material properties. **Cognitive skills** are about reasoning and creativity - engaging in hypothesis-driven modifications and proposing novel structures.

LLMs for material discovery would primarily benefit researchers at the cognitive stage. Several works such as CrystaLLM, MatterGPT, and AtomGPT (Antunes et al., 2024; Chen et al., 2024; Choudhary, 2024) aim to leverage the generative capability of LLMs to generate useful and novel CIFs. These works however, are fundamentally limited by the scope of pretraining data - generations are plausible mostly only for data-rich domains of ideal, bulk crystals. The hypothesis which motivates our work is that LLMs could generate useful CIFs past data-rich domains if the more tractable problem of CIF modification is solved as opposed to ab initio generation. Learning CIF modification is learning *how* rather than *what* to generate, requiring both motor and perceptual skills as a prerequisite. In current literature, perceptual skills have been tested through question-answer (QA) style benchmarks e.g. LLM4Mat-Bench (Niyongabo Rubungo et al., 2025), but less attention has been given to testing motor skills. To address this gap, our research question asks: how can we measure and improve LLM "crystallographic motor skills", i.e. ability to manipulate atoms in crystal structures?

In this work we introduce **AtomMotor-1K**, a compact test set of 1500 questions to benchmark reasoning LLMs on CIF motor skills. The test is made of 10 CIF action types, broken down from the real-world structural modifications which researchers may perform on CIFs. To the best of our knowledge, this is the first benchmark which examines this fundamental skill of crystallography in LLMs. The dataset of AtomMotor-1K was generated from our **AtomWorld data generator**, which can generate unlimited samples for each action, and also be used to support LLM training. We used AtomMotor-1K to evaluate several frontier text-based models, which through algorithmic approaches were generally able to succeed at simpler tasks such as adding or moving atoms, but struggled with more complex tasks such as rotating around an atom. Notably, some actions intuitive to humans had high error rates, while others considered more tedious were solved unexpectedly well. Furthermore, for the analysis of AtomMotor-1K we designed a series of tests (PointWorld, CIF-Gen, CIF-Repair, Chemical Competence Score, Struct-Prop) to isolate different aspects of LLM weaknesses and explore the relation that motor skills tests have with perceptual and cognitive skills.

We expect AtomWorld and AtomMotor-1K to gain increasing relevance as LLMs improve. LLMs have traditionally struggled with the spacial reasoning and long syntax following skills required by our benchmark - but this may soon change with rapid advancements in tool-augmented design (Hu et al., 2025), diffusion LLMs (Nie et al., 2025; Song et al., 2025), and as language-aligned video generation (Zheng et al., 2024; DeepMind, 2025) and robotics (Assran et al., 2025) models become increasingly capable. LLMs capability to manipulate crystal geometries is an important yet underexplored topic, and we believe the AtomWorld playground can play a foundational role in both testing and developing this in tomorrow's LLMs.

## 2 RELATED WORK

**LLMs for crystallography.**   LLMs have been primarily explored for their capabilities in CIF generation and QA. LLMs have been demonstrated to hold an innate ability to generate crystal structures when pretrained on millions of CIF files (Antunes et al., 2024). This process may be further reinforced through evolutionary search frameworks (Gan et al., 2025). However, as LLMs are pattern predictors, the search space is fundamentally limited by the scope of the pretraining data. LLMs can also be instruction fine-tuned to predict crystal properties or provide general QA responses from CIF, e.g. AlchemBERT, NatureLM, Darwin 1.5, etc (Liu et al., 2025; Xia et al., 2025; Xie et al., 2025; Van Herck et al., 2025; Nate Gruver & Ulissi, 2024). In data modelling, MatText (Alampara et al., 2025) investigates if such QA can be improved through different textual representations. Crystallography QA is well benchmarked, with the most comprehensive being LLM4Mat-Bench (Niyongabo Rubungo et al., 2025), consisting of approximately 2 million composition-structure-description pairs. Crystallography benchmarks also cover multimodal LLMs, e.g. work by Polat et al. (2025) investigates the generation of structural annotations to crystallographic images.

Tool-augmented LLMs such as OSDA Agent (Hu et al., 2025) improve structure generation through coupling computational chemistry tools to LLMs. These tool-augmented design frameworks are able to address the lack of in-depth chemistry knowledge of LLMs without expensive (and not always effective) fine-tuning. LLMs may be able to reliably handle geometric CIF modification through tool-augmentation.

**Multimodal reasoning.**   Approaches such as multimodal chain-of-thought (Multimodal-CoT) and visualization-of-thought (VoT) (Zhang et al., 2024; Wu et al., 2024) add image modalities to the reasoning trace rather than pure textual chain-of-thought. In particular, Multimodal-CoT with under 1 billion parameters achieved state of the art in state-of-the-art performance on the ScienceQA benchmark, outperforming larger models like GPT-3.5. As CIF describes a 3D challenge, these results suggest that multimodal reasoning approaches can be highly applicable to improving LLM ability on CIF geometry tasks, as well as reasoning-intensive QA and structure generation/modification tasks. Approaches to multimodal representation may also be influenced from developments in video generation and robotics, where models such as Genie 3 and V-JEPA 2 (DeepMind, 2025; Assran et al., 2025) are increasingly capable of understanding real-world physics and integrating this with natural language input/output. Finally, with the training objective of diffusion LLMs (Nie et al., 2025; Song et al., 2025) to be noise reversal, they have an advantage in understanding structural text compared to autoregressive LLMs - with LLaDA (Nie et al., 2025) surpassing GPT-4o in a reversal poem completion task. This also suggests diffusion LLMs may be inherently capable of

differentiating between valid and invalid modifications to CIF - important for geometric modification tasks. Developments in multimodal reasoning and diffusion suggest that LLMs may be on the cusp of being able to grasp the 3D CIF environment, making it important to benchmark this progress.

# 3 PLAYGROUND DESIGN: ATOMWORLD

## 3.1 ATOMWORLD GENERATOR

The dataset of AtomMotor-1K was generated from the scalable AtomWorld data generator. The data follows a three-part structure: two CIF files of "before" and "after" states, and an action prompt describing the change - with the goal of the LLM to yield the "after" state, given the "before" state and action. A flowchart describing the workflow from data generator to benchmark is presented in Figure 1.

The CIF format contains many optional and extensible fields. Using arbitrarily mixed CIF variants would introduce additional sources of uncertainty that are not directly related to the manipulation abilities we intend to evaluate. For this reason, we adopt the default CIF representations generated from *pymatgen* (Ong et al., 2013) and the Materials Project (MP) (Jain et al., 2013) as a standardized input format for all tasks. We leave studying how LLM performance varies across different CIF styles, as well as across other structure formats such as POSCAR and XYZ for future work.

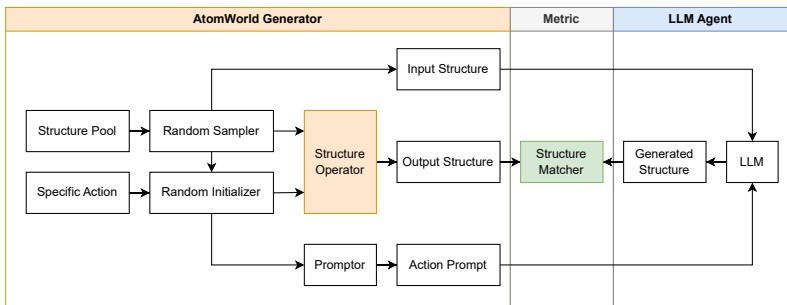

Figure 1: AtomWorld benchmark flowchart. The AtomWorld generator follows a structured data flow: the random sampler selects a structure from a predefined structure pool; the random initializer parametrizes the chosen action template by assigning atom indices and/or positions; the structure operator applies the instantiated action to the original structure to obtain the target structure; and the prompter generates a natural language description aligned with the action. The resulting (input structure, action prompt) pairs are then fed into the LLM agent system, whose generated structure is compared against the target structure using the `StructureMatcher` from *pymatgen* to compute the desired evaluation metric.

All actions currently supported by AtomWorld are detailed in Table 1. These actions are designed to be translatable into the real-world structural modifications which researchers may perform, e.g.:

- Point defect & Doping: `change, remove, add, insert_between, swap`
- Surface generation: `delete_below`
- Structure perturbation: `move, move_towards, rotate_around`
- Supercell creation: `super_cell`

## 3.2 ATOMWORLD FOR LLM TRAINING

The AtomWorld playground can be used to generate data suitable for LLM training, for instance the three-part structure of CIF-before + Action Prompt to CIF-after could feed directly into LLM pretraining. Alternatively, the same evaluation metric for AtomWorld benchmark could be used as the learning reward for reinforcement learning (RL). We leave LLM training for future work.

Table 1: Actions and the corresponding Action Prompt for AtomWorld.

| Action name | Action prompt |
| --- | --- |
| change | Change the atom at index {index} into {new_symbol} in the cif file. The indices of atoms are started from 0. |
| remove | Remove the atom at index {index} from the cif file. The indices of atoms are started from 0. |
| add | Add one {symbol} atom at the Cartesian coordinate {position} to the cif file. |
| move | Move the atom at index {index} by {d_pos} angstrom in the cif file. |
| move_towards | Move the atom at index {index1} towards the atom at index {index2} by {distance} angstrom in the cif file. |
| insert_between | Insert a {symbol} atom in the line between atoms at indices {index1} and {index2}, and the inserted atom must be {distance:.2f} angstrom from atom at {index1} in the cif file. |
| swap | Swap the spatial positions of atoms at indices {index1} and {index2} in the cif file. The indices of atoms are started from 0. |
| delete_below | Delete all atoms whose z coordinate is lower than the atom at index {index} in the cif file. Excluding itself and atoms with the same z coordinate. |
| rotate_around | Rotate all surrounding atoms within {radius} angstrom of the center atom at index {index} by {angle} degree around the axis {axis} in the cif file. The rotation should following the right-hand rule. |
| super_cell | Create a supercell with the size {dim_0}x{dim_1}x{dim_2}. |

## 3.3 COMPLEMENTARY TESTS

To support the analysis of AtomMotor-1K, we design a set of complementary tests spanning format literacy, spatial reasoning, and property-oriented understanding. These tests play the role of breaking down AtomMotor-1K to systematically target different levels of reasoning, and also to tease the applicability of agentic CIF modification workflows through integrating tests for perceptual skills.

1. **PointWorld:** A stripped-down variant of AtomWorld for measuring the inherent difficulty of each geometric operation. Structures are represented as a set of points in three-dimensional space, expressed in raw coordinate format like "$[[x_1, y_1, z_1], [x_2, y_2, z_2]]$". Models are then asked to apply geometric operations directly on these points and return the transformed coordinates. This setting removes the complexities of CIF files and serves as a controlled test of whether the LLM can handle spatial transformations at all.

2. **CIF literacy tests:**

   (a) **CIF-Repair:** Evaluates whether the model can recognize and correct corrupted or incomplete CIF files, ensuring basic robustness to noisy inputs. The CIF-Repair task is designed as the most fundamental test of CIF reading ability. The test involves CIF files with common and misleading syntax errors, such as missing tags and wrong tag names, such as "_cell_length_a" being incorrectly written as "_cell_length_x". The model is expected to correct these errors and produce a valid CIF file. A full list of corruptions is illustrated in Appendix A.6

   (b) **CIF-Gen:** Evaluates whether the model can explicitly produce syntactically valid CIFs for simple prototype crystals (e.g., sc, fcc, bcc, perovskite), thereby examining familiarity with CIF conventions and basic materials knowledge (as opposed to the open-ended CIF generation explored by Antunes et al. (2024); Chen et al. (2024); Choudhary (2024)).

   (c) **Chemical Competence Score (CCS):** This test assesses a model's latent chemical knowledge by evaluating its precision in distinguishing chemically accurate from inaccurate descriptions of crystal structures. While this test is a "perceptual skills" test, we use this to measure the effect that chemistry pretraining has on LLM performance in "motor skills" tasks. Following the methodology of Bran et al. (2025), the dataset was constructed by sampling 600 unique crystal structures from the Materials Project, with corresponding descriptions generated using Robocrystallographer (Ganose & Jain,

2019). An inaccurate dataset was then created by replacing one sentence in each original description with a sentence describing a different crystal. Because the CCS is computed from the token log-likelihoods at the model's final layer, access to these probabilities is required; this score can be calculated only for locally-run models.

3. **StructProp:** Highlights the deeper challenge of connecting crystal structures with their associated properties. Since properties are determined by structure. This task is not pursued here as a systematic benchmark. Instead, we include StructProp to **underscore the importance of structural understanding as a prerequisite for materials design**, pointing toward the longer-term goal of enabling LLMs to reason about structure-property relationships. For the Struct-Prop task, a model is required to perform actions on a given structure to achieve a desired change direction in a specific property.

## 4 EXPERIMENTAL SETUP

### 4.1 MODELS AND PARAMETER RANGES

**LLMs evaluated:** Gemini 2.5 Pro, GPT-o3, GPT-o4-mini, Deepseek Chat, Llama-3 70B, and Qwen-3 (4B, 8B, 14B, 32B).

Our selection of LLMs covers frontier closed models and strong open-source baselines. We chose the Qwen-3 series to test for parameter scaling effects. We also considered science-specialised LLMs; e.g. NatureLM (Xia et al., 2025), and MatterChat (Tang et al., 2025). However, these were excluded due to either unable to produce outputs in the required CIF format, or not currently accessible via public APIs or code implementations.

For tool-augmented LLMs, we designed a preliminary framework that enables interaction with tools such as Pymatgen. Although this workflow still requires refinement and future iterations may yield stronger results, the current findings already illustrate meaningful trends.

### 4.2 EVALUATION PROTOCOL

Our evaluation is focused on reasoning LLMs. No additional fine-tuning or reinforcement learning was performed. Inference was run with default API parameters. The prompt templates used for all tests can be found in Appendix A.4.

### 4.3 DATASETS

1. **AtomWorld: AtomMotor-1K.** The AtomWorld data generator can in principle produce an unbounded number of test cases. **AtomMotor-1K** is a representative set of 1500 questions. It contains 5×250 questions for actions add, move, move_towards,

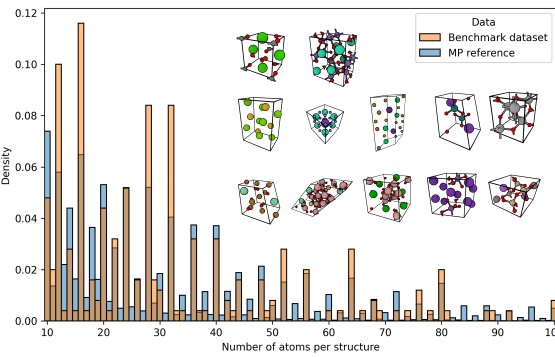

Figure 2: Distribution of the "before" material structures used in AtomMotor-1K (total of 250 structures) by their number of atoms. Also depicted for reference is MP's distribution and visualisations of some structures used in AtomMotor-1K.

`insert_between`, and `rotate_around`; and $5 \times 50$ questions for actions `remove`, `change`, `swap`, `delete_below`, and `super_cell`. The CIFs used for "before" states are consistent across action classes as a control. A distribution of these structures by their size is depicted in Figure 2. For the `super_cell` action, the output structure was specified to range from 2 to $8 \times$ the original cell size. An example of an `insert_between` test is illustrated in Appendix A.5.

2. **PointWorld.** Implemented four AtomWorld-analogous action types - `move`, `move towards`, `insert_between`, `rotate_around`. Only two points are implemented in one sample, to make the task more fundamental. For each action, 250 samples were tested on Deepseek V3, and 50 samples on Gemini 2.5 Pro. This relatively limited test sample was enough to indicate the pattern of task difficulty in AtomWorld.

3. **CIF-Repair and CIF-Gen.** 22 generated samples for CIF-Repair and 20 manually-labelled samples for CIF-Gen across all LLMs used in AtomWorld. We used only a small scale of tests to isolate the LLM's understanding of CIF syntax and material structure representation from the demands of AtomWorld tasks.

4. **CCS.** 600 crystal structure descriptions and their corresponding corrupted versions were generated using Robocrystallographer. As only open-source models (Llama-3 70B and Qwen3 series) were tested, the full dataset could be evaluated without the cost constraints of closed-source APIs. This dataset serves to isolate a picture of each model's latent understanding of crystal structures in natural language.

5. **StructProp.** 209 manually-labelled structures are collected according to Strukturbericht type (Mehl et al., 2017). Due to the testing cost of DFT calculation pipelines, we choose 10 samples to test - for each LLM used in AtomWorld, for each property (band gap and bulk modulus). This was enough to give an indication of how effective LLMs could be for hypothesis-driven CIF modification.

### 4.4 METRICS

1. **Success rate.** Used for all datasets except CCS. Defined as the number of test cases successfully pass all of the following errors divided by the total number of test cases. These errors are categorized into three hierarchical levels:

   (a) **Wrong output format.** The LLM's response must enclose the generated structure within a predefined tag so that it can be correctly extracted from the textual output. Failure to do so constitutes an output format error.

   (b) **Wrong structure format.** Even if the structure is successfully extracted, its file format may still be invalid or incompatible with downstream processing tools. Such cases are counted as structure format errors.

   (c) **Mismatch of structures.** For structurally valid outputs, we compare them with the target structures using `StructureMatcher` with a site tolerance of 0.5. Any generated structure whose site matching exceeds this tolerance is considered a mismatch.

2. **Success rate (StructProp).** The success metric for StructProp includes two additional criterion: whether the generated structure can be used in first principle calculations, and whether the modified structure fulfills the correct property change. A success rate of over 50% for a model indicates the model does better than random guessing.

3. **Mean maximum distance** (`max_dist`). Used for AtomWorld, PointWorld, CIF-Gen datasets. Computed only for structurally valid outputs that pass the tolerance check. For each matched pair of structures, we calculate the maximum pairwise atomic displacement after optimal alignment, and then average this value across all test cases. The `max_dist` metric is used because it is generally more significant than the RMSD value in our cases. This is because only a few or even a single atom is "moved" while others remain unchanged, making the maximum displacement a more representative indicator of the structural difference.

4. **CCS score.** This metric was used to evaluate whether LLMs could discern between correct and incorrect crystal structure descriptions. The underlying assumption is that models with a stronger understanding of crystal structures will assign higher likelihoods in their final layer to correct statements than to incorrect ones. Accordingly, the metric measures the separation between the distributions of mean ranks for correct and incorrect descriptions.

We report this separation using Cohen's *d* effect size, where larger values indicate a clearer distinction between the two distributions and, by extension, a stronger ability of the model to recognise correct statements based on the provided structure and its surrounding context.

# 5 RESULTS

## 5.1 ATOMWORLD: ATOMMOTOR-1K

The main results of AtomMotor-1K, alongside complementary tests are presented in Figure 3.a. We see some separation of the AtomWorld actions into easy (`change`, `remove`, `swap`, `add`), moderate (`move`, `move towards`, `insert between`) and hard difficulty (`delete below`, `rotate around`) levels based on their success rates. The moderate and hard difficulty tasks constitute greater requirements of multi-step or spatial reasoning. We also notice the mean `max_dist` metric increase for the more difficult tasks (minus tasks not requiring structural perturbations). The `max_dist` distributions for each task are shown in Appendix B.3. One interesting finding is that , the `super_cell` task cannot be well categorised into these difficulty tiers as the success rates range from Llama3-70B's 4% to 98% from GPT-o3 - it's both easy (just large-scale repetition) and difficult (requires long-context output) at the same time. The parameter scaling results in Figure 3.c and d illustrate that larger models generally achieve higher success rates and smaller displacements. However, with improvements with scale being marginal with more difficult tasks, and noting that Qwen3-32B outperformes Llama3-70B across most tasks, it suggests that architectural design and training strategies play an equally important role as parameter size. We note that the failure cases appear largely random, with outputs ranging from unmodified input structures to partially modified or slightly perturbed atom positions. A detailed mechanistic analysis would require extensive human inspection and is left for future work.

Our evaluation of our tool-augmented LLM framework on AtomWorld tasks found noticeable gains in model performance. However, the gains are somewhat limited, particularly for more complex actions. Detailed results and comprehensive analysis can be found in Appendix C.

## 5.2 POINTWORLD AND CIF LITERACY TESTS

**PointWorld.** These results are listed in Table 2. Both models are able to reliably output a parseable output with near perfect "success rate" (errors in Deepseek V3 model at `insert_between` tasks are due to it sometimes attempting to write Python scripts instead of performing the calculation). The indicator of task difficulty is in the mean `max_dist` scores, where models performed well on `move`, `move_towards`, and `insert_between`, but found `rotate_around` significantly more difficult. The former actions could be solved with straightforward numerical calculations (e.g., addition or weighted averaging), which LLMs can handle reliably. In contrast, models often attempted to compute a rotation matrix for the `rotate_around` action and failed to apply it consistently, leading to high mean `max_dist`.

Table 2: Model performances on simplified point-based tasks. Success rate (Succ. rate) indicates the ratio of unreadable outputs from LLMs. Mean `max_dist` is calculated by the maximum distance between generated and target points after Hungarian sort.

| Action | Gemini 2.5 Pro (50 frames) | | Deepseek V3-0324 (250 frames) | |
|---|---|---|---|---|
| | Succ. rate (%) | mean `max_dist` (Å) | Succ. rate | mean `max_dist` |
| `move` | 100.00 | 0.0000 | 100.00 | 0.0000 |
| `move_towards` | 98.00 | 0.0045 | 100.00 | 0.3172 |
| `insert_between` | 100.00 | 0.0051 | 78.8 | 0.0642 |
| `rotate_around` | 98.00 | 16.168 | 100.00 | 14.058 |

**CIF-Repair.** These evaluations are presented in the main results of Figure 3.a. Most models were able to demonstrate a strong foundational capability in understanding CIF format and errors, with success rates of over 90%. While Llama3 and Qwen3 series have success rates falling below 60%, this

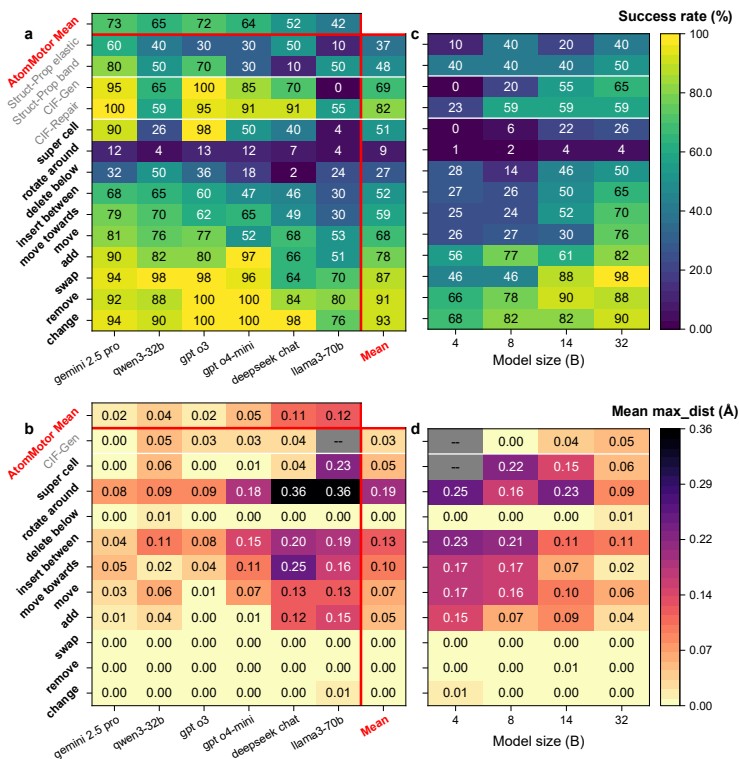

Figure 3: **a.** Success rate metric across AtomMotor-1K, CIF-Repair, CIF-Gen and StructProp datasets. **b.** Mean `max_dist` metric across AtomMotor-1K and CIF-Gen datasets. **c, d.** Parameter scaling results on Qwen3 series. The right side are some randomly sampled structures from the tested data.

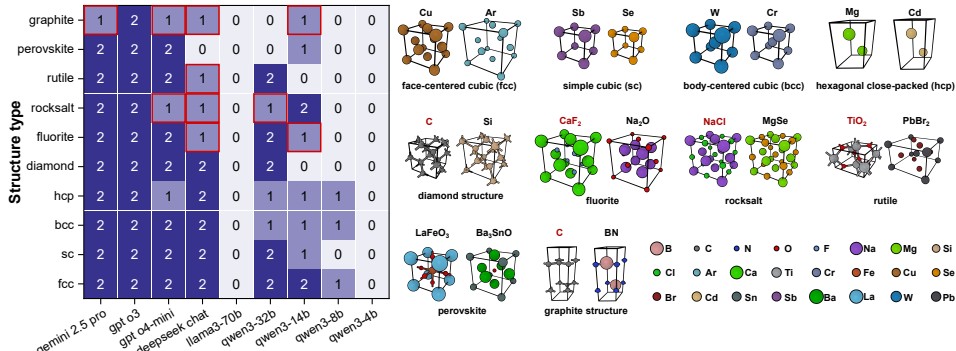

Figure 4: Visualised results of CIF-Gen task. The left side shows the number of correctly generated CIFs for each structure type. The squares marked in red indicate cases where the single correct generation is the standard prototype. The right side shows the specific 3D crystal structures for each type, where the chemical compositions in red represent the standard prototypes.

does not seem to limit their capability to yield higher success rates even in moderately challenging AtomWorld tasks.

**CIF-Gen.** These evaluations are presented in the main results of Figure 3.a and show a similar trend to CIF-Repair tasks. A closer look at the error cases in Figure 4 find that chemical compositions that define standard prototypes are generated correctly more often than non-standard compounds that crystallize in the same prototypes (e.g. NaCl vs. MgSe for rocksalt, $CaF_2$ vs. $Na_2O$ for fluorite). The fact that assymetries in training data affect LLMs in this way demonstrates that they rely more on

memorization of specific examples rather than understanding the underlying structural principles. Nevertheless, Gemini 2.5 Pro and o3 were able to demonstrate this understanding with success rates of 95% and 100%, respectively.

**CCS.** The resulting scores are reported in Table 3. Similar to AtomWorld, scaling within the Qwen3 series yielded incrementally higher scores, indicating that larger models of the same architectural design acquire a more nuanced grasp of crystal structure properties from their underlying compositions. Notably, while larger Qwen models generally perform better, the Qwen3-32B model surpasses the larger Llama3-70B, mirroring the pattern observed in AtomWorld.

Table 3: CCS score of open-source models

| Model | CCS |
|---|---|
| Qwen3 4B | 0.768 |
| Qwen3 8B | 0.829 |
| Qwen3 14B | 1.061 |
| Qwen3 32B | **1.141** |
| Llama3 70b | 0.987 |

## 5.3 STRUCTPROP

Table 4: Excerpts from Gemini 2.5 Pro reasoning trace during successful completions of StructProp tasks. Correct and incorrect identifications are highlighted.

| Structure & Goal | Core thinking process |
|---|---|
| PtS (mp-288) 

 Band gap ↓ | ... a potent strategy is to raise the energy of the valence band maximum. This can be achieved by substituting the sulfur atoms with a heavier, less electronegative element from the same group, such as Selenium (Se). The Se 4p orbitals are higher in energy than the S 3p orbitals. |
| $Ga_2S_3$ (mp-539) 

 Band gap ↑ | ... its band gap is largely determined by the electronegativity difference and bond strength between the Gallium (Ga) cation and the Sulfur (S) anion. To increase the band gap, a modification that strengthens the chemical bonds and increases the material's ionicity is required. |
| $CdAs_2$ (mp-471) 

 Bulk modulus ↑ | ... Cd is a relatively large and electropositive element, leading to bonds with significant ionic character that are less stiff than purely covalent bonds. 
 ... Zn is directly above Cadmium in Group 12 of the periodic table. It has a smaller atomic radius and is more electronegative, which leads to the formation of shorter, stronger, and more covalent bonds with As. |

These evaluations are presented in the main results of Figure 3.a. Most LLMs were generally unable to get over 50% success rate in these tasks. With the strongest performing model Gemini 2.5 Pro achieving an average success rate of 70%, we list three examples of its reasoning trace in Table 4. These examples highlight LLM knowledge of the definitions of target properties and an ability suggest plausible modification strategies, but also underlines a limited understanding of the underlying electronic structure. In the PtS case, the model correctly identified the key driver of the band gap change as the higher energy of Se 4p compared to S 3p orbitals, but stopped short of

a deeper discussion of orbital overlap and covalency - Pt-S bonding is likely to be more covalent than Pt-Se, potentially leading to additional band gap narrowing. In the $Ga_2S_3$ case, the model captured the correct trend in terms of electronegativity differences and bond ionicity. The $CdAs_2$ case highlights an incorrect reasoning flow that still lead to successful completion of the task. The model mischaracterised the relative electronegativities of Cd (1.69) and Zn (1.65), attributing the improvement to enhanced ionicity - the true effect is likely linked to stronger covalent bonding due to Zn 3d-As 3p interactions.

## 6  DISCUSSION

Isolated tests of PointWorld suggest that LLMs could perform near-perfectly for the simplified `move`, `move_towards`, and `insert_between` tasks - which suggests that the moderate (50-80%) success rate for the AtomWorld analogous tasks is due to difficulty with CIF syntax following as opposed to spatial reasoning. Yet isolated tests for CIF literacy generally found success rates of over 80%, suggesting the opposite - spatial reasoning is more difficult than CIF syntax following. The reality is likely that the task difficulty was compounded when both spatial reasoning and syntax following requirements were combined. Moreover, real-world materials modelling workflows rarely involve single-step actions as in AtomWorld. Instead, they require executing chains of operations. For example, creating supercells is often a prerequisite for other tasks: studying defect properties at a given concentration demands first generating a supercell of appropriate size before atoms can be removed or substituted. A complex instruction such as "generate defect at x% concentration" would thus entail an extended reasoning chain, amplifying the difficulty. Stronger RL specific to AtomWorld tasks could be a solution to helping LLMs understand reasoning chains relevant to CIF modification.

The AtomWorld benchmark is an essential first step: if LLMs cannot reliably perform these basic operations, it will be difficult to envision progress toward more complex materials research workflows. At the same time, solving AtomWorld does not necessarily mean relying on LLMs alone. In practice, difficult actions such as `rotate_around` are better handled by crystallography tools, and future agentic workflows will likely combine LLMs with tool support or multi-modal inputs. Our StructProp results further suggest that current LLMs already show some ability to connect structures with their properties, hinting at the feasibility of gradually scaffolding more complex reasoning within tool-augmented frameworks. From this perspective, AtomWorld can be viewed not as an end in itself but as a foundational stepping stone toward practical, full-cycle agentic materials discovery.

## 7  CONCLUSION

In this paper, we presented AtomWorld as the first benchmark that evaluates LLM motor skills in crystallography. In general, we found that chat models took an algorithmic approach to solving the geometric tasks of our benchmark. With this approach, simpler operations such as `add` could be performed more consistently, whereas more spatially demanding manipulations, particularly rotations, remain highly challenging. These tasks can be solved manually via crystallography software, but for LLMs are an important first stage to enabling higher value tasks such as developing an agentic material discovery workflow. At the same time, preliminary tests suggest that simply equipping LLMs with code tools and RAG is not sufficient to solve problems perfectly. More thoughtful toolflow design and post-training are still necessary for practical application.

LLMs have traditionally struggled with spatial reasoning tasks. However, this may be soon to change with recent developments in tool-augmented design, diffusion, video generation, and language-aligned robotics models (Hu et al., 2025; Song et al., 2025; DeepMind, 2025; Assran et al., 2025). We hope that our AtomWorld playground can play a foundational role in helping researchers of tomorrow test LLMs' understanding of 3D CIF environments.

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
