## A  ATOMWORLD SETUP DETAILS

### A.1  DATASET DISTRIBUTION

Herein, we analyze the distributions of structures collected in our benchmark in terms of elemental composition, number of atoms, number of elements, and space groups, and compare them with those from the Materials Project. The data are shown in Figure 5, Figure 6, Figure 7, and 8, respectively. The structures in our dataset are randomly sampled from the Materials Project, with the number of atoms ranging from 10 to 100. Note that our data generator is not limited to the sampled subset; in principle, any structure file can be used for data generation, and it is capable of automatically generating an unlimited amount of benchmarking data.

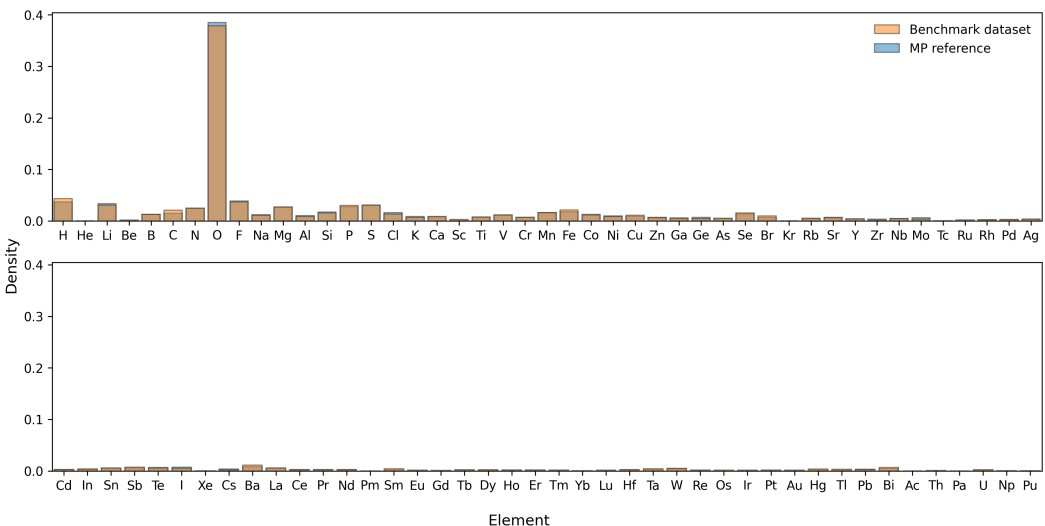

Figure 5: Distribution of structures by elemental composition, compared with the Materials Project. The orange bars represent the data used in the AtomWorld Bench, while the blue bars represent those of the Materials Project.

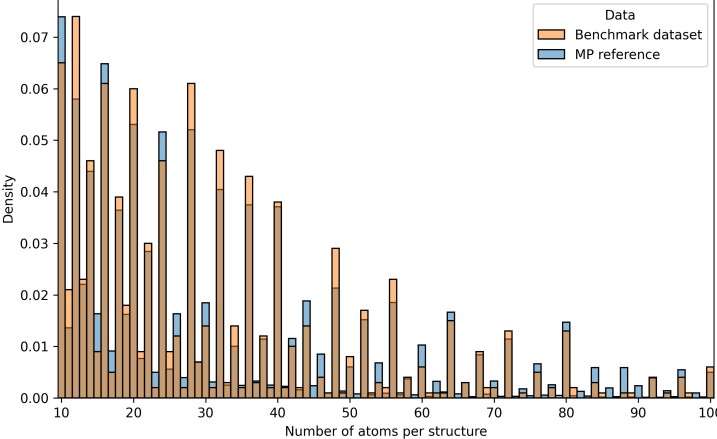

Figure 6: Distribution of structures by the number of atoms, ranging from 10 to 100, compared with the Materials Project.

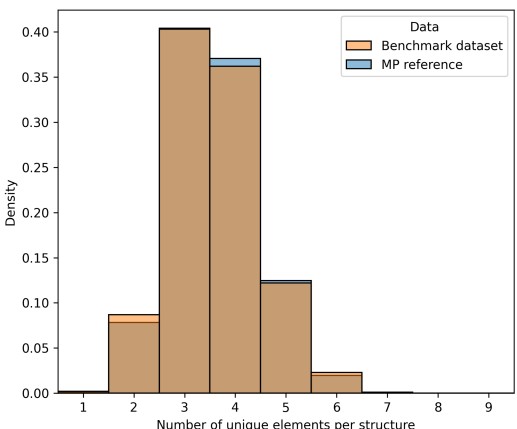

Figure 7: Distribution of the number of elements per structure compared with the Materials Project.

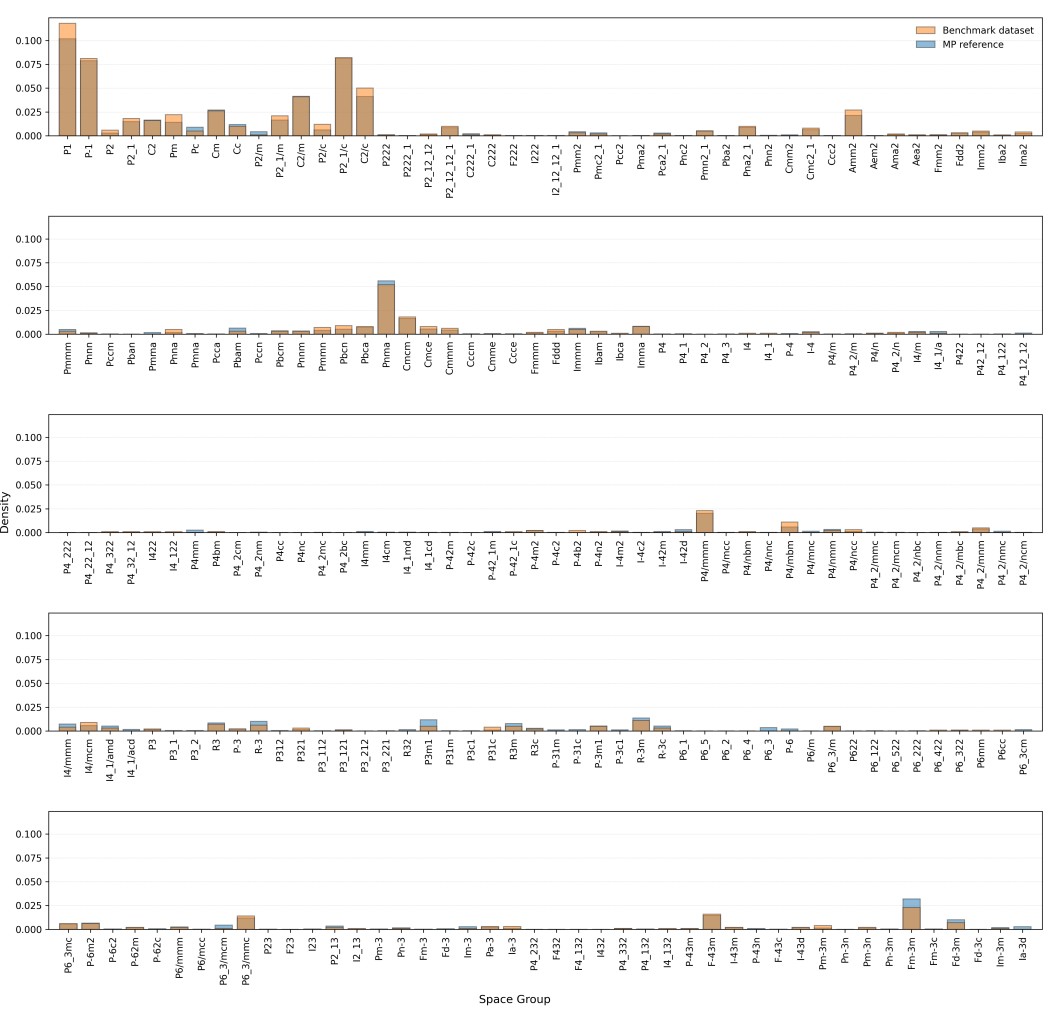

Figure 8: Distribution of structures by space group, compared with the Materials Project.

## A.2 Data generator parameters for each action

Each action is associated with a set of parameters that specify which atoms the action targets, as well as the magnitude and manner in which the action is applied. Our data generator takes a given structure and a specified action type as input, and produces a set of randomly initialized parameters that define the action. The generator then applies the action to the structure, producing the resulting data. Table 5 lists the parameters required for each action used in this work, along with the ranges from which the data generator randomly samples their initial values.

Table 5: Parameter ranges for random actions in the data generator. The input structure contains $N$ atoms, and the lattice matrix is $A = (\boldsymbol{a}_1, \boldsymbol{a}_2, \boldsymbol{a}_3)$.

| Action | Sampling ranges of parameters |
|---|---|
| change | index: $[0, N)$ |
| | symbol: $\{H, He, \ldots, Os\}$ |
| remove | index: $[0, N)$ |
| add | position: $A\boldsymbol{u}, \boldsymbol{u} \in [0, 1)^3$ |
| | symbol: $\{H, He, \ldots, Os\}$ |
| move | index: $[0, N)$ |
| | displacement: $\mathcal{N}(0, \sigma^2 I_3), \ \sigma = 2$ |
| move_towards | index1, index2: $[0, N)$, index1 $\neq$ index2 |
| | distance: $[0.1, 3)$ Å |
| insert_between | index1, index2: $[0, N)$, index1 $\neq$ index2 |
| | symbol: $\{H, He, \ldots, Os\}$ |
| | distance_ratio: $[0.1, 0.9)$ |
| swap | index1, index2: $[0, N)$, index1 $\neq$ index2 |
| | Only between atoms with different symbols. |
| delete_below | index: $[0, N)$ |
| | include_self: $\{True, False\}$ |
| rotate_around | index: $[0, N)$ |
| | radius: $[1.0, 4.0)$ Å, capped by the structure size. |
| | angle: $[45°, 315°)$ |
| | axis: $\{\pm\hat{\boldsymbol{x}}, \pm\hat{\boldsymbol{y}}, \pm\hat{\boldsymbol{z}}\}$ |
| super_cell | size: $(a, b, c) \in \{1, 2, 3, 4\}^3, a \times b \times c \leq 8, (a, b, c) \neq (1, 1, 1)$ |

## A.3 SUPPORTED ACTION PROMPTS FOR POINTWORLD.

Table 6: Examples of actions and the corresponding action prompts for point-based tasks.

| Action name | Action prompt |
|---|---|
| move | Move the point at index {index} by displacement {displacement}. |
| move_towards | Move the point at index {from_index} towards the point at index {to_index} by {distance}. |
| insert_between | Insert a new point between points at indices {index1} and {index2}, {distance} units away from point {index1}. |
| rotate_around | Rotate all points by {angle_deg} degrees around the axis {axis}, with the point at index {center_index} as the center of rotation. The rotation follows the right-hand rule. |

## A.4 FULL PROMPT TEMPLATES

Listing 1: A prompt example for a specific task of AtomWorld

```
You are a CIF operation assistant. You will be given an input CIF content
 and an action prompt. Your task is to apply the action described in the
action prompt to the initial CIF content. The coordinates in the action
are in Cartesian format. Return the modified CIF content in cif format
within <cif> and </cif> tags.

Please ensure the output is a valid CIF file, with correct formula, and
atom positions.

Input CIF content:
{The specific CIF file is inserted here}

Action prompt: Insert Lu between atoms at indices 6 and 5 that is 4.03
angstrom from atom 6.
```

Listing 2: A prompt example for the PointWorld task

```
You are a spatial reasoning expert. You will be given an initial set of
points and an action prompt describing an operation on these points. The
final modified points after applying the action must be returned inside <
answer> and </answer> tags. The format inside the tags must exactly match
 the input points format. All indices are zero-based. Please ensure the
answer inside <answer> and </answer> tags is parseable and strictly
formatted.
Initial points data:
{coordinate_array},
Action prompt:
{action_prompt},
```

Listing 3: A prompt example for CIF-repair tasks

```
You are a CIF operation assistant. You will be given a CIF content that
may be corrupted or incomplete. Your task is to examine the CIF content
and fix any issues to ensure it is a valid CIF file. If there are missing
 values that cannot be repaired directly, you can use the [
VALUE_TO_BE_INSERTED] as hints to fill in the missing values. Please
ensure the output is a correct CIF file. Return the fixed CIF content
within <cif> and </cif> tags. Input CIF content:

{broken_cif}
```

Listing 4: A prompt example for a CIF-gen task about perovskite structure

```
You are a materials science expert. Please generate some simple and
standard structures in the CIF format according to the requirements. You
must strictly follow the CIF format specifications. Since the symmetry-
related information can be complex, please write the CIF file with P1
symmetry. Please ensure the output is a correct CIF file. Return the
fixed CIF content within <cif> and </cif> tags.

Requirements:

Please generate a CIF file for {formula} with a {structure_type}
structure, according to the following information about the convensional
cell:

- Lattice constant a: {lattice_constant_a}
- The {center_atom} atom is at the center of the octahedron formed by
surrounding atoms.
```

Listing 5: A prompt example for StructProp tasks

```
You are a material design expert. Your task is to modify a given CIF file
 to achieve a desired change in a specific material property. Please
analyze the given CIF file and the target property. Identify the key
structural features and elemental composition that influence the
specified property. Propose a specific modification to the structure.
This modification must be one or a combination of the following:
 1. Element Substitution;
 2. Lattice Parameter Adjustment;
 3. Atomic Coordinate Adjustment.
 Please ensure the output is a correct CIF file. Return the modified CIF
 content within <cif> and </cif> tags.
 Input CIF content:
 {The specific CIF file is inserted here}

 Your goal: modify the CIF file accordingly to {target_trend} the {
 target_property}.
```

## A.5 ILLUSTRATIVE EXAMPLE OF THE FRAMEWORK

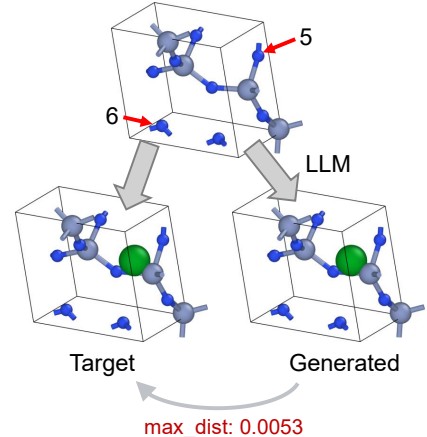

Figure 9: The workflow of a specific `insert_between` task.

To provide a concrete understanding of our proposed AtomWorld Bench, we present an illustrative example of its workflow. This case study focuses on a specific task: inserting a Lu atom between the fifth and the sixth atoms in the specific CIF structure. The prompt used here is listed in Appendix A.4.

The workflow randomly selects the atom indices and determines the position of the atom to be inserted based on the selected atoms. Based on the initialized action, the framework gives out a target structure. The LLM will also generate a structure after processing the prompt, as shown in Figure 9. In this example, the two structures are nearly identical, with a `max_dist` of 0.0053 Å, indicating high accuracy.

## A.6 LOGIC ON GENERATING CIF-REPAIR TASK

To systematically evaluate LLM performance on CIF repair, we constructed a set of partially corrupted CIFs via two types of operations:

1. **Removal of essential lines:** Certain CIF fields are critical for correct structure parsing. The essential tags include:
   - `_cell_length_a`, `_cell_length_b`, `_cell_length_c`
   - `_cell_angle_alpha`, `_cell_angle_beta`, `_cell_angle_gamma`
   - `_atom_site_type_symbol`, `_atom_site_label`, `_atom_site_symmetry_multiplicity`
   - `_atom_site_fract_x`, `_atom_site_fract_y`, `_atom_site_fract_z`
   - `_atom_site_occupancy`

2. **Replacement of essential tags with misleading variants:** Instead of random typos, tags are systematically replaced with misleading but syntactically valid alternatives. Examples of mappings include:
   - Change the `a`, `b`, `c` into `x`, `y`, `z`; `u`, `v`, `w` or `i`, `j`, `k`.
   - Change the `x`, `y`, `z` into `a`, `b`, `c`; `u`, `v`, `w` or `i`, `j`, `k`.
   - Change `_atom_site` string into `_atom`.
   - Change `_cell` string into `_lattice`.
   - Change `_cell_length` and `_cell_angle` strings into `_cell`.

## A.7 DFT COMPUTATION DETAILS

All density functional theory (DFT) calculations, including band gap and bulk modulus evaluations, were performed using the Vienna Ab initio Simulation Package (VASP) with the projector-augmented wave (PAW) method (Kresse & Hafner, 1993; Kresse & Furthmüller, 1996a;b; Kresse & Joubert, 1999) and the PBEsol exchange–correlation functional(Perdew et al., 2008). High-throughput workflows for both properties were automated using the atomate2 package (Ganose et al., 2025). Unless otherwise specified, calculation parameters followed the default settings in atomate2. Example calculation scripts are provided in the github repository.

For the band gap calculations, a k-point mesh with a grid density of 100 Å$^{-3}$ was employed, and electronic self-consistency was converged to $10^{-5}$ eV. The band gap was extracted from the uniform k-point calculation stage. For the bulk modulus calculations, a plane-wave energy cutoff of 600 eV and a k-point grid density of 400 Å$^{-3}$ were used. Total energy and ionic relaxations were converged to $10^{-6}$ eV and 0.01 eV/Å, respectively, to balance computational cost and accuracy. In the initial relaxation stage, Gaussian smearing with $\sigma = 0.05$ eV was applied, while in the deformation stage the tetrahedron method was adopted for Brillouin zone integration.

## B SUPPLEMENTARY EXPERIMENTAL DATA

### B.1 PROMPT CHOICE AND MISINTERPRETATION

During tests, we found that certain prompt formulations could cause LLMs to misinterpret spatial actions as text-level editing, specifically for the `swap` action (Listing 6). Prompts that focused on textual "indices" without explicitly framing the task as a spatial transformation caused some models to attempt CIF-text rewriting instead of manipulating atomic coordinates. To verify whether this was an intrinsic ambiguity in the wording, we asked three domain experts in materials modelling to perform the same tasks using only the less explicit prompt. Both interpreted the action correctly and did not exhibit the LLM-style misinterpretation. As domain experts with long-term experience in atomic modelling, their prior exposure to structure manipulation likely provides an additional intuition for identifying the spatially intended reading, rather than a purely textual one. In contrast, LLMs appear to exhibit a default "interpretive bias" toward textual position unless the spatial nature of the task is made explicit.

Our goal, however, was not to optimize prompts, but to ensure consistent and unambiguous task interpretation. All experiments therefore use a single unified prompt set, chosen simply to remove obvious sources of misunderstanding while maintaining comparability across tasks. These prompts are likely not globally optimal but effectively prevent semantic confusion without engaging in extensive prompt tuning.

Listing 6: Less explicit and explicit spatial prompts for `swap` action

```
Less explicit prompt:
"Swap atoms at indices {self.index1} and {self.index2} in the cif file.
The indices of atoms are started from 0."

Success rate (Deepseek-chat):  22%
Success rate (Qwen3 32B):  50%

Explicit spatial prompt:
"Swap the spatial positions of atoms at indices {self.index1} and {self.
index2} in the cif file. The indices of atoms are started from 0."

Success rate (Deepseek-chat):  64%
Success rate (Qwen3 32B):  98%
```

### B.2 PERFORMANCE SENSITIVITY ANALYSIS

#### B.2.1 SENSITIVITY ON THE NUMBER OF ATOMS

To quantitatively investigate the influence of the number of atoms on the action success rate, we prepared test data that is as clean as possible, using a single action with consistent and simple action parameters. For the structures, we employed different supercell sizes of the same base structure. For each structure, we have tested 10 times. In this case, we chose the `insert_between` action to insert a Hydrogen atom in the middle between atoms at indices 3 and 5. We chose BiOF (mp-762304, $P2_1/c$, 12 atoms per unit cell) as the test system, and generated supercells with 12, 24, 48, 96, 144, and 216 atoms. This system was selected to provide moderate structural complexity while being representative of a three-element compound with ample data in the Materials Project.

As shown in Figure 10, the success rates of all tested models decrease as the number of atoms in the CIF increases. The larger model (Qwen3 32B) shows a slower decline compared to the smaller ones (Qwen3 4B). Beyond 200 atoms, all three models failed in every test case. This trend suggests that while larger models are more robust to increasing input size, extremely large structures still exceed the models' effective reasoning capacity.

#### B.2.2 SENSITIVITY ON BRAVAIS LATTICE TYPES

We also investigated the effect of structural symmetry on the action success rate. Given that there are 230 space groups in total, analyzing all of them would be impractical. Therefore, we focused on the

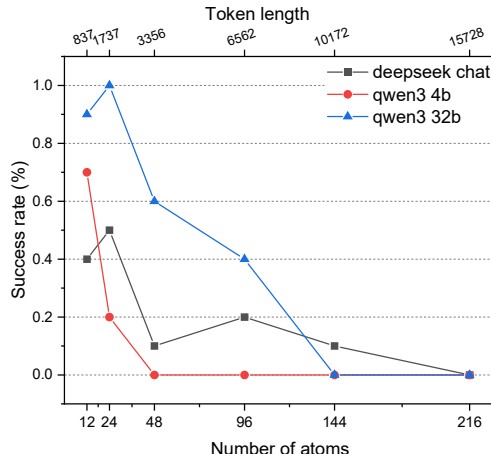

Figure 10: The relation between success rate and the number of atoms (directly related to token lengths).

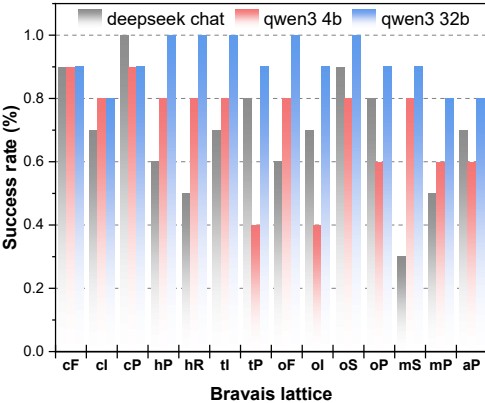

Figure 11: The success rate under the 14 Bravais lattice types.

14 Bravais lattices for statistical analysis. To minimize the influence of other factors, we performed random sampling from the Materials Project, selecting conventional cells containing 12 to 28 atoms, and collected 10 structures for each lattice type. As shown in Figure 11, the Bravais lattices are arranged roughly in decreasing order of symmetry, from face-centered cubic (cF) to simple primitive (aP). The success rates of the three tested models show no clear correlation with the lattice symmetry. However, the models tend to achieve slightly lower accuracy on low-symmetry systems, although a definitive conclusion would require more detailed investigation.

### B.2.3 SENSITIVITY ON THE ACTION POSITIONS

We further evaluated whether the position of the target atom in the CIF sequence affects the action success rate. To this end, we prepared paired test cases using the same structure, where the target atoms appeared either early or late in the CIF atom list. All other factors were kept fixed. Specifically, we used the `insert_between` action with a distance_ratio of 0.45 and fixed the inserted atom type as Hydrogen. For the "early" setting, the selected atom indices were 0 and 1, whereas for the "late" setting, the last two indices in the CIF were used. The underlying structures are identical to those used in the standard `insert_between` tasks.

As shown in Table 7, the tested models consistently achieve higher success rates when the target atoms appear earlier in the CIF sequence. Although the difference is not large, the trend is observed across the two evaluated models. Overall, the results indicate that atom ordering has some impact on performance, but the effect remains limited under our testing conditions. To overcome this issue, we believe that the future agentic systems should use specific tools or modalities to understand the structures, then perform the actions.

Table 7: The success rate for different action positions.

| Action position | Deepseek-chat (%) | Qwen3 32B (%) |
|---|---|---|
| Index 0, 1 | 90 | 96 |
| Index -2, -1 | 82 | 80 |

### B.2.4 PERFORMANCE ON MOVING ALL ATOMS

We further evaluated the `move_all` action, which requires the model to translate the entire structure by a specified displacement vector. As summarized in Table 8, the success rates of all tested models drop substantially compared to the single-atom `move` tasks. In many failed cases, the generated structures show noticeable randomness in the atomic positions, indicating that the models have difficulty performing consistent global translations.

Unlike the other main metrics in AtomWorld, evaluating `move_all` poses an additional challenge: a rigid translation preserves the structure up to translational symmetry. As a result, the standard `StructureMatcher` cannot be directly applied because it intentionally factors out translational degrees of freedom. To quantify deviations, we implemented a custom metric similar to `StructureMatcher` but without enforcing translational invariance, comparing atoms by their index correspondence and computing the resulting coordinate deviations.

Table 8: The metrics for the `move_all` action.

| Models | Success rate (%) | mean `max_dist` (Å) |
|---|---|---|
| Deepseek-chat | 54 | 0.1111 |
| Qwen3 4B | 12 | 0.1118 |
| Qwen3 8B | 24 | 0.0252 |
| Qwen3 14B | 40 | 0.0467 |
| Qwen3 32B | 46 | 0.0487 |

### B.3 THE `MAX_DIST` VIOLIN PLOTS

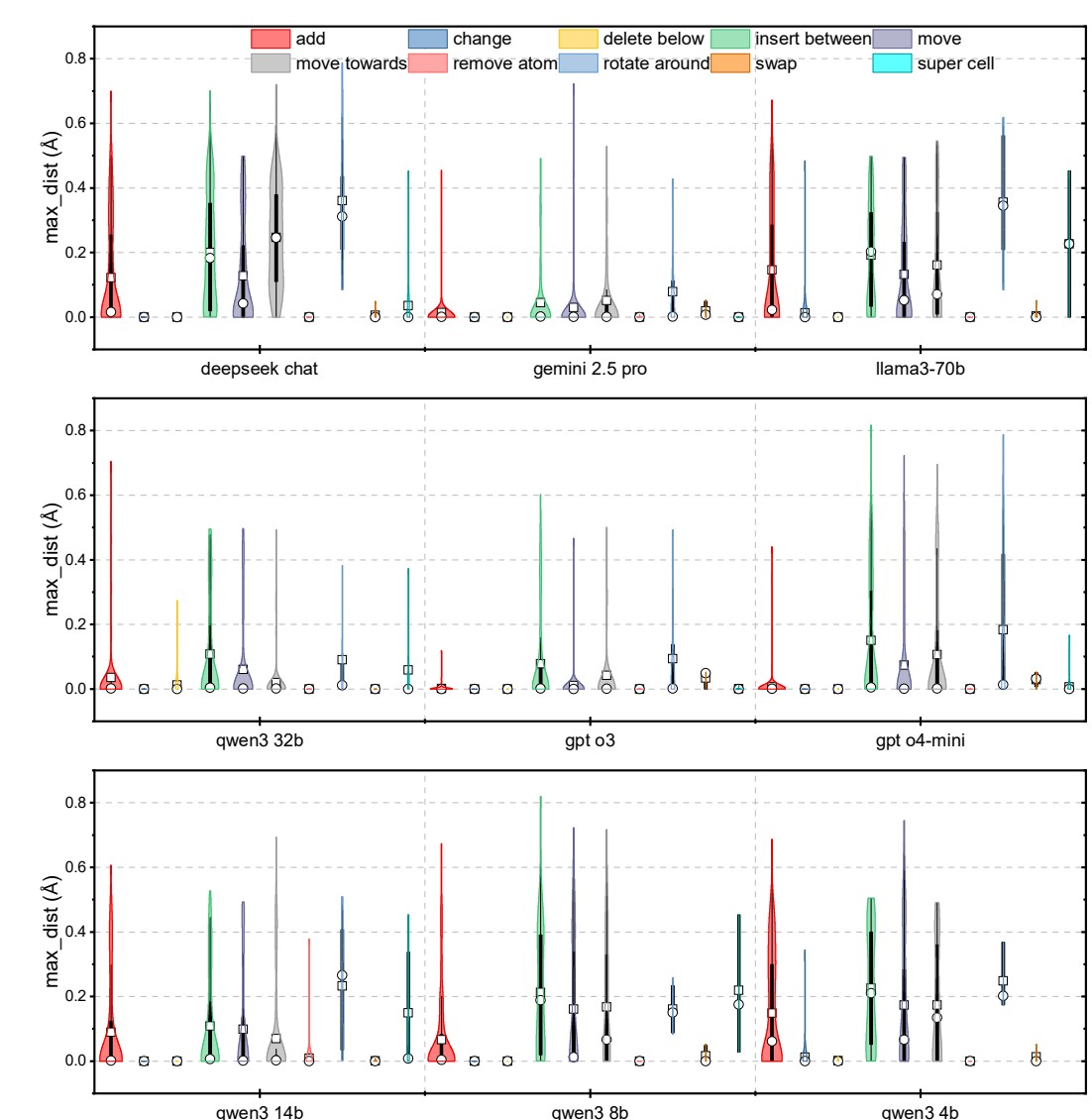

Figure 12: The violin plots of `max_dist` of evaluation results. The hollow squares indicate the mean values, and the hollow circles indicate the medians.

## C  EVALUATIONS OF TOOL AUGMENTED LLM FOR ATOMWORLD

**System Design**    As shown in Figure 13, we adopt a code generation-based approach to accomplish structural operations. This process is divided into two steps: first, we perform RAG-based retrieval over the pymatgen library to obtain relevant APIs; second, we conduct code generation to complete the user-specified action.

**Knowledge Graph Retrieval (RAG)**    The first step of our pipeline is to retrieve relevant pymatgen APIs using RAG. We leverage the code-graph-rag project(Liu et al., 2024) to extract structured information from the codebase and build a knowledge graph in Memgraph, where nodes represent code entities such as modules, classes, methods, and fields, and edges capture relationships like inheritance and usage. The retrieval process is orchestrated by a primary LLM, implemented using Deepseek-chat, which performs task decomposition, reasoning, and tool invocation. Specifically, the translator LLM, also implemented with Deepseek-chat, is used as a tool by the primary LLM

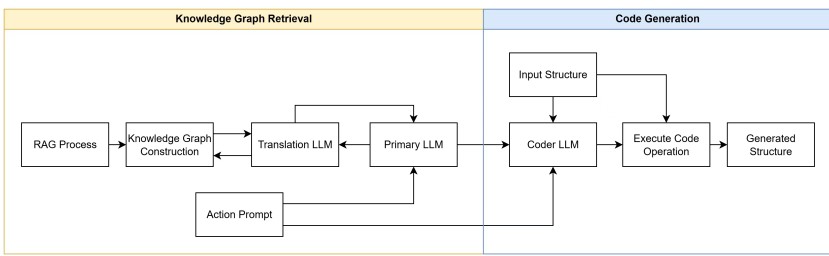

Figure 13: The flowchart for the code generation-based approach for the AtomWorld benchmark tests.

to convert natural language queries into graph queries. The output of this process is a JSON file containing relevant pymatgen APIs, which is later used to guide code generation.

**Code Generation**    Code generation is performed using Deepseek-chat, conditioned on the input CIF file, the user action prompt, and the APIs retrieved from the RAG stage. The system strictly follows the retrieved API signatures to ensure correctness and prevent hallucination. The generated Python code is then executed together with the input CIF file to produce the modified crystal structure.

Table 9: Comparison of model performances between Deepseek-chat with and without tools.

| Action | With tools | | Without tools | |
| --- | --- | --- | --- | --- |
| | Succ. rate (%) | mean `max_dist` (Å) | Succ. rate | mean `max_dist` |
| `remove` | **100.0** | 0.0000 | 84.0 | 0.0000 |
| `insert_between` | **83.0** | 0.0076 | 45.6 | 0.2004 |
| `rotate_around` | **18.0** | 0.1648 | 6.8 | 0.2561 |

As evident from Table 9, incorporating retrieval-augmented generation (RAG) and structure manipulation tools significantly improves the model's performance across the tested actions. The `remove` action, which is relatively straightforward, achieves a perfect success rate of 100%. However, more complex actions, such as `insert_between` and `rotate_around`, still present challenges. The success rate for `insert_between` is 83%, with some errors remaining, while `rotate_around` demonstrates a relatively low success rate of 18%.

These findings highlight a key insight: while the integration of RAG tools and coding ability facilitates substantial improvements in model performance, further refinements are crucial to fully address the real-world requirements of structural modification tasks. Specifically, additional task-specific fine-tuning or reinforcement learning is necessary to enhance the model's robustness, particularly for more complex structural operations. Future work will focus on these aspects to ensure more reliable and scalable applications.