# OpenReview forum: "AtomWorld: Benchmarking Spatial Reasoning in Large Language Models on Crystalline Materials"
_ICLR.cc/2026/Conference — Submitted to ICLR 2026_

### Official Review · Reviewer_4UhP · 2025-10-30

**Soundness:** 3
**Presentation:** 3
**Contribution:** 3
**Rating:** 6
**Confidence:** 4

**Summary:**

The paper presents the AtomWorld benchmark to evaluate LLMs on tasks based on Crystallographic Information Files (CIFs), including structural editing, CIF perception, and property-guided modeling. The authors built this standardized benchmark to systematically evaluate the core reasoning abilities of LLMs across diverse atomic structures, hence bridging the gap for applying LLMs for materials science domains. The benchmarking results reveal that current LLMs consistently fail in structural understanding and spatial reasoning, frequently making errors on structure modification tasks, and even in basic CIF format understandings, potentially leading to cumulative errors in subsequent analysis. As the future goal, AtomBench is expected to play a foundational role in both testing and developing the understanding of 3D CIF environments in the next generation of LLMs.

**Strengths:**

1. The authors built the AtomWorld, which lays the ground for advancing LLMs toward robust atomic-scale modeling. In contrast, previous work mainly focused on text-based or property-prediction tasks; this benchmark fills a major gap in assessing spatial understanding, which is essential for advancing agentic materials discovery.

1. Beyond AtomBench, the authors add complementary tests, namely PointWorld, CIF-Repair, CIF-Gen, Chemical Competence Score (CCS), and StructProp, each of which targets a different level of reasoning. This design provides a systematic view of where LLMs fail: syntax vs. materials reasoning.

1. The authors provide a few insightful discussions: distinguish between spatial reasoning and following CIF syntax; identify that combining both of them significantly increases difficulty; and point out that tool-augmented and multimodal approaches are needed for progress.

**Weaknesses:**

1. The actions are designed to resemble the real-world structural modifications; however, the modifications may not be consistent with material physics and chemistry. An arbitrary addition/removal/moving doesn't necessarily result in a scientifically meaningful structure. Therefore, the benchmark is more about evaluating the LLM's knowledge of CIF grammar than materials science.

1. The structures in AtomBench are sampled from the Materials Project, which may limit diversity, especially for complex or defective systems.

1. The paper identifies failures but lacks mechanistic explanations of why models fail, which may limit the insight of the work.

**Questions:**

1. For the samples per action, are they drawn from a broad range of chemistries and symmetries (e.g., metals, oxides, perovskites), or mostly from a small subset of Materials Project data entries?

1. Have the authors tested prompt robustness of LLMs? If models are highly prompt-sensitive, AtomWorld might measure instruction-following ability more than spatial reasoning.

---

> ### Author Response · Authors · 2025-11-23
>
> **Part 1:**
>
> Thank you for your feedback and for pointing out the relevance and importance of our problem topic to materials science. We clarify your queries along with our responses below:
>
> **Q1:** The modifications may not be consistent with material physics and chemistry. An arbitrary addition/removal/moving doesn't necessarily result in a scientifically meaningful structure. Therefore, the benchmark is more about evaluating the LLM's knowledge of CIF grammar than materials science. **(For Weakness 1)**
>
> **A1:** Thank you for your suggestion. We emphasize that structural manipulation is a prerequisite for physically valid modeling. Just as a coder must learn syntax before writing efficient algorithms, LLMs must master different spatial operations before addressing thermodynamic stability. The current version of AtomWorld targets this foundational capability gap.
>
> **Q2:** The structures are sampled from the Materials Project, which may limit diversity, especially for complex or defective systems. **(For W2)**
>
> **A2:** Thank you for raising this. We understand this concern. Structural diversity of our base/starting CIFs matters to our benchmark as it changes the task difficulty in the same way that larger numbers would change the difficulty of an arithmetic problem. In our revised manuscript (Appendix B.2) we have included an exploration into the factors that make the starting CIFs more difficult and found that it is largely tied to size of the CIF (i.e. number of atoms in the structure) rather than its symmetries. Therefore structure diversity is not a limitation of our benchmark since it is not the primary impact on task difficulty currently.
>
>
> **Q3:** The paper identifies failures but lacks mechanistic explanations of why models fail, which may limit the insight of the work. **(For W3)**
>
> **A3:** Thank you for pointing this out. Mechanistic explanations for closed-weights models (e.g., GPT-4, Claude) are fundamentally constrained by lack of access to internal states. AtomWorld contributes by providing behavioral insights and error modes, which is the standard for evaluating frontier models where architectural details are proprietary.
>
> **Q4:** For the samples per action, are they drawn from a broad range of chemistries and symmetries or mostly from a small subset of Materials Project data entries? **(For Question 1)**
>
> **A4:** The structures are sampled randomly from the Materials Project database, which covers similar data distributions across the number of atoms, element species, and space groups. To clearly show them, we have drawn several figures comparing our data distribution with those in the MP, as shown in Appendix A.1.  We have also added a detailed dataset distribution for AtomMotor-1K in Appendix A.1 of the revised manuscript, and we also provide a brief explanation in the main text to help clarify how CIFs are sampled in Section 4.3.
>
> In brief, the structures in our dataset are randomly sampled from the Materials Project, with the number of atoms ranging from 10 to 100, which covers the similar element, number of atoms, and space group diversity to the MP database, which is sufficient to this benchmark.

---

> ### Author Response · Authors · 2025-11-29
>
> **Part 2:**
>
> **Q5:** Have the authors tested prompt robustness of LLMs? If models are highly prompt-sensitive, AtomWorld might measure instruction-following ability more than spatial reasoning. **(For Q2)**
>
> **A5:** We thank the reviewer for raising this valuable point. We have observed that certain prompt formulations can lead to misinterpretation by LLMs, particularly when the spatial meaning is implicit rather than explicitly stated. For instance, an earlier version of the delete_below prompt (“delete all the atoms below …”) frequently caused models to remove the latter part of the CIF file instead of the atoms that are spatially lower. This issue was resolved once the prompt was clarified to “delete all the atoms whose z coordinate is lower than …”.
>
> We also found that the LLM’s misunderstanding of prompts is not strictly identical to human experts (added in Appendix B.1). In brief, for the swap actions, when using “Swap atoms at …” as actions, LLMs can generally fail and just exchange the lines’ order rather than their spatial positions, unless using “Swap the spatial position of …”, while human experts tend to give correct answers even without explicitly given this spatial description. This is likely due to the insight from long-term experience in atomic modelling of domain experts.
>
> In general, we find that as long as the prompt contains explicit spatial descriptions, LLMs do not fundamentally misunderstand the task and typically produce at least a semantically consistent output, irrespective of correctness. Our objective is not prompt optimization, but to ensure consistent and unambiguous task interpretation. Therefore we treat prompts as a controlled variable, with all experiments using a single unified prompt set, chosen simply to remove obvious sources of misunderstanding while maintaining comparability across tasks. These prompts are likely not globally optimal but effectively prevent semantic confusion without engaging in extensive prompt tuning.
>
> We appreciate your valuable feedback. If you have any further suggestions or questions, we welcome you to raise them for discussion.

---

### Official Review · Reviewer_HUw1 · 2025-10-31

**Soundness:** 3
**Presentation:** 3
**Contribution:** 2
**Rating:** 4
**Confidence:** 4

**Summary:**

This paper introduces Atom World, a benchmark for evaluating the spatial reasoning capabilities of Large Language Models (LLMs) in the context of atomic structures. The benchmark tests various spatial transformations (move, swap, rotate) on structures represented in CIF (Crystallographic Information File) format. While the problem is relevant, the paper has methodological gaps and presents results that may contradict prior work in the field.

**Strengths:**

- Addresses an important question about LLMs' spatial reasoning capabilities in scientific domains
- Several thoughtful design choices in benchmark construction

**Weaknesses:**

### 1. Missing Discussion of Prior Work

There has been prior work on the abilities to model geometric properties with LLM. One of them is [MatText](https://arxiv.org/abs/2406.17295), which was followed up by a work from Andrew Gordon Wilson’s group](https://arxiv.org/abs/2404.02444).


### 2. Missing Experimental Details

For example, the "move" operation lacks essential specifications (perhaps I missed them):
- What is the distance of movement?
- How are movement distances sampled?
- Does the distribution of sampled distances correlate with error magnitudes?
- The maximum distances shown are relatively small—why?

### 3. Insufficient Analysis of Structural Factors

The paper does not examine how performance relates to:
- Number of atoms in the structure
- Structural complexity
- Symmetry properties
- Other important structural characteristics

### 4. Questionable Choice of Evaluation Metric

The current metric may not capture what matters most. For molecular/atomic structures, **preserving intramolecular distances** may be more important than successfully translating the entire structure. The benchmark should weight internal geometric consistency appropriately.

### 5. Oversimplified Treatment of CIF Format

The CIF dictionary contains numerous fields and is extensible by design. The paper presents the problem as if there is one canonical way to represent structures, glossing over significant flexibility in the format. A cleaner experimental design would:
- Compare atom-only CIF syntax vs. full CIF syntax
- Control for the number of atom types
- Control for which atom types are present
- Systematically vary these factors

(Not saying that this needs to be covered in experiments, but perhaps it should be reflected in writing).

### 6. Anthropomorphization of LLMs

The authors anthropomorphize LLMs in the introduction and discussion, referring to "thinking traces" that LLMs obviously do not have.

### 7. Puzzling and Inconsistent Results

Several results are difficult to interpret:

- **Swap performance is surprisingly poor** - this seems anomalous and lacks explanation (the paper only says “failed surprisingly”)
- **Move results may be artifacts** of how the benchmark is constructed rather than genuine capability measurements (depends on the distance the atom is being moved)
- What happens when the entire structure is moved uniformly?

As the authors themselves seem to acknowledge (“The reality is likely...”), it is hard to obtain deep insights from the current benchmark design --- what are the underlying limitations of the models that cause problems?

**Questions:**

1. How do you reconcile your conclusions with the findings of MatText and Gruver et al.?
2. Can you provide complete specifications for all transformation operations, particularly movement distances and their sampling procedures?
3. Can you provide an analysis broken down by:
   - Number of atoms
   - Structural complexity metrics
   - Symmetry groups
4. Have you considered metrics that prioritize preservation of intramolecular distances?
6. Why is swap performance so poor, and is this a genuine model limitation or an artifact?
7. What happens when you translate entire structures uniformly (rigid body translation)?

---

> ### Author Response · Authors · 2025-11-23
>
> **Part 1:**
>
> We appreciate your insightful questions and helpful suggestions. For clarity, we have organized your queries along with our responses below:
>
> **Q1:** Missing Discussion of Prior Work: There has been prior work on the abilities to model geometric properties with LLM. One of them is MatText, which was followed up by a work from Andrew Gordon Wilson’s group](https://arxiv.org/abs/2404.02444). **(For Weakness 1 and Question 1)**
>
> **A1:** We appreciate the reviewer bringing MatText to our attention. While MatText investigates how modelling material structures through textual representations can improve QA performance on the materials, it does not address the generative modification of crystal structures (CIFs), which is our contribution of work. Our approach goes beyond textual description to directly manipulate structural files. We have updated the manuscript in Section 2 Related Work to explicitly contextualize our work against MatText.
>
> **Q2:** Missing Experimental Details For example, the "move" operation lacks essential specifications (perhaps I missed them): What is the distance of movement? How are movement distances sampled? Does the distribution of sampled distances correlate with error magnitudes? The maximum distances shown are relatively small—why? **(For W2 and Q2)**
>
> **A2:** Thank you for raising this. We have added detailed specifications of the data generator's parameters to Appendix A.2. of the revised manuscript. The parameters are listed here for convenience:
>
> Table 1: Parameter ranges for random actions in the data generator. The input structure contains N atoms, and the lattice matrix is  A.
> | Action | Sampling ranges of parameters |
> |-------|-------|
> | `change` | `index`: [0, N); `symbol`: {H, He, …, Os} |
> | `remove` | `index`: [0, N) |
> | `add` | `position`: A·u, u∈[0,1)³; `symbol`: {H, He, …, Os} |
> | `move` | `index`: [0, N); `displacement`: 𝒩(0,σ²I₃), σ=2 |
> | `move_towards` | `index1`, `index2`: [0, N), index1≠index2; `distance`: [0.1, 3) Å |
> | `insert_between` | `index1`, `index2`: [0, N), index1≠index2; `symbol`: {H, He, …, Os}; `distance_ratio`: [0.1, 0.9) |
> | `swap` | `index1`, `index2`: [0, N), index1≠index2; Only between atoms with different symbols |
> | `delete_below` | `index`: [0, N); `include_self`: {True, False} |
> | `rotate_around` | `index`: [0, N); `radius`: [1.0, 4.0) Å, capped by the structure size; `angle`: [45°, 315°); `axis`: {±x̂, ±ŷ, ±ẑ} |
> | `super_cell` | `size`: (a,b,c)∈{1,2,3,4}³, a×b×c≤8, (a,b,c)≠(1,1,1) |
>
> There might be some misunderstanding of the mean max_dist here, which is not the total distance the action “moved” the structure, but a metric to measure the maximum mismatch between the LLMs’ output structure and the correct target structure. This metric can only be applied to the output that can correctly “match” the target structure under site_tolarence=0.5 condition. So it is naturally bound below a “small quantity”.
>
> **Q3:** Insufficient Analysis of Structural Factors: The paper does not examine how performance relates to: Number of atoms in the structure; Structural complexity; Symmetry properties; Other important structural characteristics. **(For W3 and Q3)**
>
> **A3:** We thank the reviewer for this insightful suggestion regarding failure analysis. We have conducted a comprehensive evaluation, now detailed in Appendix B.2 of the review.
>
> 1. System size and symmetry: We observe an inverse correlation between success rate and system size(atom count/token length), a trend consistent with generation difficulty in larger systems. However, we do not find a clear relation between Bravais lattice type and the success rate.
>
> 2. Target Position: Our analysis suggests that there is only a marginal correlation with the action’s target atom position and the success rate.
>
> 3. Structural complexity, we respectfully note that no widely accepted quantitative metric exists across different material systems. Complexity is generally influenced by atom count and symmetry;
>
> In summary,  we used the number of atoms and symmetry considerations as the most robust proxies. We believe these effectively capture the difficulty of the generation task, though we remain open to testing specific alternative metrics if the reviewer has a particular recommendation.

---

> ### Author Response · Authors · 2025-11-29
>
> **Part 2:**
>
> **Q4:** Questionable Choice of Evaluation Metric: The current metric may not capture what matters most. For molecular/atomic structures, preserving intramolecular distances may be more important than successfully translating the entire structure. The benchmark should weight internal geometric consistency appropriately. **(For W4 and Q4)**
>
> **A4:** We clarify that the primary objective of our benchmark is to evaluate atomic-level structural operations- the ability of an LLM to accurately execute specific spatial and syntactic instructions rather than to enforce absolute chemical or physical correctness (e.g.,bond lengths or intramolecular distances). While geometric preservation is critical for de novo generation, it is orthogonal to the goal of instruction following. A model must first demonstrate the motor skill to manipulate coordinates according to instructions before optimizing for physical validity. Therefore, our metric correctly prioritizes operational accuracy.
>
> **Q5:** Oversimplified Treatment of CIF Format: The CIF dictionary contains numerous fields and is extensible by design. The paper presents the problem as if there is one canonical way to represent structures, glossing over significant flexibility in the format. A cleaner experimental design would: Compare atom-only CIF syntax vs. full CIF syntax; Control for the number of atom types; Control for which atom types are present; Systematically vary these factors. (Not saying that this needs to be covered in experiments, but perhaps it should be reflected in writing). **(For W5)**
>
> **A5:** We appreciate your insightful comment. In our benchmark, we intentionally adopt a simplified and standardized CIF subset generated from pymatgen in order to maintain comparability across action types and to isolate the spatial-reasoning aspect without introducing additional confounding factors arising from variability in CIF syntax. Your suggestion points to a valuable direction for future development, to extend AtomWorld to include additional CIF variants, as well as other widely used structure formats such as POSCAR and XYZ. We leave these for future work. To make this clear, we have also added a clarification to explain this format issue in Section 3.1 of the revised manuscript.
>
> **Q6:** Anthropomorphism of LLMs: The authors anthropomorphize LLMs in the introduction and discussion, referring to "thinking traces" that LLMs obviously do not have. **(For W6)**
>
> **A6:** Thank you for raising this, but we respectfully disagree with this. We refer to “thinking traces” as synonymous to the “reasoning” or “chain-of-thought” tokens prior to the output of the CIF answer. The analysis of “thinking traces” is helpful to uncover the chemical knowledge of the models or to investigate how incorrect CIFs were outputted. We have not attributed emotions, intentions, or self-agency to LLMs.

---

> ### Author Response · Authors · 2025-11-29
>
> **Part 3:**
>
> **Q7:** Puzzling and inconsistent results: Several results are difficult to interpret:
> - Swap performance is surprisingly poor - this seems anomalous and lacks explanation (the paper only says “failed surprisingly”)
> - Move results may be artifacts of how the benchmark is constructed rather than genuine capability measurements (depends on the distance the atom is being moved)
> - What happens when the entire structure is moved uniformly?
>
> As the authors themselves seem to acknowledge (“The reality is likely...”), it is hard to obtain deep insights from the current benchmark design --- what are the underlying limitations of the models that cause problems?  **(For W7, Q5 and Q6)**
>
> **A7:** Swap performance: We appreciate you highlighting the anomalous performance of the swap action. We started by asking three human domain experts to solve the original swap questions to examine whether there were any aspects that could be misunderstood by humans. All of them answered correctly, indicating that humans naturally resolve the latent ambiguity. In contrast, the consistent error patterns observed in initial LLM results suggest that models often defaulted to a purely textual interpretation of “swap”, leading to no substantive geometric changes. This indicates that the ambiguity arises primarily from divergent reasoning assumptions between humans and LLMs about what a valid “swap” entails. Motivated by this finding, we refined the prompts by incorporating explicit spatial descriptions (e.g., “swap the [spatial positions] of …”) and retested the LLMs, and found that LLMs performed better and the success rate increased from 22% to 64% for Deepseek-chat. We have updated our swap action results with the updated prompts accordingly. Nevertheless, we emphasise that the focus of our work is not prompt optimization - but for instances where prompt quality is poor enough to affect the validity of our benchmark, it is valuable to be addressed. We have added a discussion about the prompt choices to Appendix B.1 of the revised manuscript. We have also double-checked the reasoning steps of LLM outputs for other tasks to make sure prompt ambiguity is not an issue.
>
> Move results may be artifacts: Given that there may have been some misunderstanding regarding the 'move' in the results, such as the maximum distance, we would like to first gain a clearer understanding of whether the 'artifact' you are concerned about still exists, or whether it has become more specific. If you still maintain that the suggested experiment is essential for a thorough evaluation, we are prepared to perform the requested analysis and include the results in the next revision.
>
> Uniform move: This was an interesting problem, and we have benchmarked open source models on this action.
>
> Table 2: The metrics for the move_all action.
>
> | Models          | Success rate (%) | mean `max_dist` (Å) |
> |----------|---------|-----------|
> | Deepseek-chat   | 54              | 0.1111              |
> | Qwen3 4B        | 12              | 0.1118              |
> | Qwen3 8B        | 24              | 0.0252              |
> | Qwen3 14B       | 40              | 0.0467              |
> | Qwen3 32B       | 46              | 0.0487              |
>
> In general, success rates are less than 60%, dropping substantially compared to the single-atom move action. The output wrong structures are generally showing noticeable randomness, indicating that the models have difficulty performing consistent global translations. The details and experiment setups have been included in Appendix B.2.4 of the revised manuscript.
>
> We appreciate your comprehensive feedback. We welcome any further suggestions or questions for discussion.

---

### Official Review · Reviewer_RAHg · 2025-10-31

**Soundness:** 2
**Presentation:** 1
**Contribution:** 2
**Rating:** 0
**Confidence:** 5

**Summary:**

This paper introduces AtomWorld, a benchmark designed to assess large language models (LLMs) on spatial reasoning tasks involving crystalline materials. The benchmark focuses on atomic-level manipulations defined in Crystallographic Information Files (CIFs), encompassing ten operations such as atom addition, rotation, substitution, and supercell construction. The authors also design complementary tasks (PointWorld, CIF-Gen, CIF-Repair, Chemical Competence Score, StructProp) to isolate reasoning versus syntax-following abilities. Experiments across GPT, Gemini, Llama, DeepSeek, and Qwen models demonstrate that current LLMs can handle simple operations but struggle with 3D geometric reasoning.

**Strengths:**

The benchmark pipeline is implemented correctly, and the evaluation metrics (success rate, mean max distance) are well-defined. Includes PointWorld, CIF-Repair, CIF-Gen, and StructProp, offering a multidimensional view of LLM capability. Applies to multiple frontier models (Gemini, GPT-o4, DeepSeek, Llama, Qwen). The framework could support reinforcement learning or tool-augmented evaluation in future.

**Weaknesses:**

Only ~10K samples despite millions of public CIFs available (e.g., COD, Materials Cloud, OQMD, NOMAD). This limits generality and robustness. Claims of LLM “spatial reasoning failure” are not quantified beyond accuracy; no correlation analysis between task difficulty and model architecture/training. No uncertainty estimates, variance reporting, or ablation analysis. Related Work misplaced after Discussion; results buried within tables and dense text.

While the paper is motivated by the goal of testing spatial understanding in scientific LLMs, the benchmark’s data scale and challenge level are limited — only ~10K samples drawn from a small subset of the Materials Project, despite the availability of millions of publicly accessible CIFs (e.g., NOMAD, OQMD, AFLOW, and COD).

As a result, the benchmark does not meaningfully probe the limits of current models. Most models perform moderately well on all but the hardest operations, suggesting the dataset may lack sufficient complexity or novelty to reveal qualitative differences in spatial reasoning. The organization of the paper also makes comprehension difficult: the Related Work section appears after Discussion and before Conclusion, breaking logical flow.

Below papers can be considered for comparison:
https://arxiv.org/abs/2312.00111
https://openreview.net/forum?id=Q2PNocDcp6
https://arxiv.org/abs/2506.13051

**Questions:**

Why was the dataset limited to ~10K CIF actions when large-scale CIF repositories (e.g., NOMAD, OQMD, COD) contain millions of structures?

How are CIFs sampled—are they balanced across crystal systems and chemistries?

Would scaling AtomWorld to include defects, polymorphs, or molecular crystals yield a more meaningful challenge?

Have you compared the benchmark’s discriminative power (e.g., ability to separate model tiers) to datasets like ChemBench (arXiv:2506.13051)?

Why does the paper place Related Work after Discussion? Could you restructure to better contextualize contributions earlier?

Have you explored tool-augmented baselines (e.g., with ASE or Pymatgen API access) to distinguish reasoning from symbolic manipulation errors?

---

> ### Author Response · Authors · 2025-11-23
>
> **Part 1:**
>
> We appreciate your insightful questions and helpful suggestions. For clarity, we have organized your queries along with our responses below:
>
> **Q1:** Only ~10K samples despite millions of public CIFs available (e.g., COD, Materials Cloud, OQMD, NOMAD). This limits generality and robustness. **(For Weakness 1 and Question 1)**
>
> **A1:** We thank the reviewer for highlighting the availability of large-scale repositories like COD and NOMAD. However, our intention is not to construct a materials-diversity dataset, but to build a **motor-intelligence benchmark** that evaluates an LLM’s ability to execute **atomic-level editing operations** on crystalline structures. In our setting, the CIF serves as a textual point set on which a specific spatial manipulation is performed. The central challenge therefore lies not in exhausting materials-space coverage, but in ensuring sufficient **task diversity** and enough **structural variability** to make these manipulations non-trivial.
>
> This is analogous to **mathematical symbolic-manipulation benchmarks**:
>
> Increasing numerical magnitude may affect difficulty,but generating arbitrarily large numbers is neither required nor meaningful for testing whether a model understands the operation itself. Likewise, expanding CIF diversity to millions of entries does not increase the validity of an **operation-level reasoning benchmark**.
>
> Regarding robustness: We emphasize that a sample size of \~10,000 is statistically rigorous for an evaluation benchmark. It provides sufficient volume to yield stable error margins and distribution convergence. Scaling to millions would increase computational cost without altering the assessment of the model's reasoning mechanism.
>
> To ensure appropriate structural complexity, we intentionally sample from the Materials Project - over 150,000 high-quality, DFT-relaxed structures with broad coverage across space groups, element types, and cell sizes. This avoids experimental noise (e.g., partial occupancies, unrelaxed sites) that would act as confounding variables. The critique conflates **data scaling** (necessary for training a model) with **capability evaluation**(which requires a representative, high-fidelity sample).
>
> **Q2:** no correlation analysis between task difficulty and model architecture/training. **(For W1)**
>
> **A2:** We thank you for your suggestion to analyze correlations between task difficulty and model architecture or training details.While we agree that such an attribution analysis is scientifically desirable, it’s **fundamentally constrained by the opacity of current frontier models**. Most models evaluated in our work(e.g., GPT-o3, Gemini 2.5 pro) do not disclose their training data composition, internal architecture variants, or fine-grained training procedures. As a result, any correlation analysis would rely on unverifiable assumptions and would not meet scientific rigor. For this reason, wThis approach aligns with standard practice for benchmarking proprietary LLMs, ensuring that our reported results remain reproducible and free from conjectural bias.
>
> **Q3:** No uncertainty estimates, variance reporting, or ablation analysis. **(For W1)**
>
> **A3:** Violin plots for the max_dist metric have been already presented in Appendix B.3 (previously B.2). Since the result distribution is far from a standard Gaussian distribution, we do not directly provide the variance, but the distributions. We have also referenced these plots in Section 5.1 of the revised manuscript.
>
> Ablation studies are indeed essential when a paper proposes a new model or algorithmic component whose internal design choices directly affect performance.
>
> However, AtomWorld is an architecture-agnostic benchmark suite rather than a model or algorithm. As such, the benchmark does not control, nor should it modify, the internal architecture, training procedure, or data composition of the LLMs being evaluated. Ablating components of models is infeasible and outside the scope of a benchmark whose goal is to measure capabilities rather than alter them.
>
> To illustrate potential pathways for improvement, we did include an example of a partially tool-augmented setting (Appendix C), demonstrating how external crystallographic tools can enhance LLM performance. This complements our benchmark without requiring ablations on models whose internal mechanisms are inaccessible.
>
> **Q4:** Related work is misplaced after the Discussion.  **(For W1 and Q5)**
>
> **A4:** We have changed it to appear after the introduction in our revised manuscript.

---

> ### Author Response · Authors · 2025-11-29
>
> **Part 2:**
>
> **Q5:** Results are buried within tables and dense text. **(For W1)**
>
> **A5:** We would like to clarify that the primary results were already presented in compact visual form (current Figure 3), each designed specifically to make inter-model and inter-task patterns immediately apparent without requiring the reader to parse raw numbers. These figures have been the central entry point of the Results section since the original submission.
>
> To accommodate this concern, we further improved readability by adding summary rows and columns to Figure 3 and by expanding the accompanying textual description. These additions make the high-level findings visible at a single glance, even for readers who prefer not to inspect the full matrix. While we have enhanced the presentation for convenience, the results were never “buried”; they were already structured for clarity consistent with field norms.
>
> **Q6:** Benchmark does not meaningfully probe LLM limits since they performed moderately well on all but the hardest operations. Suggests lack of sufficient complexity or novelty. **(For W3)**
>
> **A6:** We respectfully point our that assessment of ‘moderate’ performance overlooks the distinct requirement of scientific automation versus standard NLP tasks.
>
> 1. **The Necessity of a difficulty gradient**: A well-calibrated benchmark must demonstrate a discriminative gradient - verifying competence on elementary operations while exposing failure modes in complex reasoning. If the models failed universally, the benchmark would be indistinguishable from noise. The fact that models succeed on simple operations but degrade on 3D geometric reasoning shows AtomWorld as a precise instrument for measuring the **frontier of capability**.
>
> 2. The ‘moderate accuracy’ trap in Science: we argue that in the context of physical science, ‘moderate accuracy’ is insufficient for real applications like autonomous agents.  Unlike creative writing, scientific workflows require high-fidelity execution. If an LLM requires a lot of human-in-the-loop or a costly DFT validation step (e.g., via atomate2) to catch frequent hallucination, the utility of LLM in physical science application is negated.
>
> Atomworld does not lack complexity; rather, it exposes a critical reality: current models have not yet crossed the reliability threshold required for true physical science application. The benchmark successfully quantifies this gap, which is exactly its intended contribution.
>
> **Q7:** How are CIFs sampled—are they balanced across crystal systems and chemistries? **(For Q2)**
>
> **A7:** We have added a detailed dataset distribution in Appendix A.1, and we also provide a brief explanation in the main text to help clarify how CIFs are sampled in Section 4.3. In brief, the structures in our dataset are randomly sampled from MP, with the number of atoms ranging from 10 to 100. The diversity of our elements, number of atoms, and space group is similar to the MP database, which is sufficient for this benchmark.
>
> **Q8:** Would scaling AtomWorld to include defects, polymorphs, or molecular crystals yield a more meaningful challenge? **(For Q3)**
>
> **A8:** We agree that defects, polymorphs and molecular crystals represent a higher tier of material complexity. However, scaling to these complexities is premature for current capabilities.
> Our analysis (Appendix B.2.2) identifies system size (atom count) as the dominant variable correlating with failure. With success rate dropping quickly for systems >100 atoms, introducing defects( which often require larger supercells) or complex molecular topologies would likely result in a **‘floor effect’** (near zero performance across all models). Therefore, AtomWorld focuses on idealized periodic structures to build a necessary baseline. We view the inclusion of defects and polymorphs not as a correction to the current benchmark, but as the logical ‘level 2’ evaluation once models demonstrate competence on the fundamental ‘level 1’ task we present.
>
> **Q9:** Have you compared the benchmark’s discriminative power (e.g., ability to separate model tiers) to datasets like ChemBench (arXiv:2506.13051)? **(For Q4)**
>
> **A9:** The discriminative power of our benchmark is depicted in Figure 3a,c showing the range of success rates vs. models. We have not compared our benchmarks’ discriminative power to ChemBench because our domains are mismatched. ChemBench focuses on chemical reasoning and reaction prediction, while ours is on structural editing in crystallography. Instead, we have done rank correlation of our motor skills tests with specifically designed complementary tests.

---

> ### Author Response · Authors · 2025-11-29
>
> **Part 3:**
>
> **Q10:** Have you explored tool-augmented baselines (e.g., with ASE or Pymatgen API access) to distinguish reasoning from symbolic manipulation errors? **(For Q6)**
>
> **A10:** Developing fully tool-augmented LLMs is non-trivial and beyond this benchmark. However, **we would like to clarify that we have already tested using a simple agentic workflow in the previous versions of the manuscript**, and the results remain included in (Appendix C.1) and discussed in Section 4.1. In general, the implementation of pymatgen tools will increase the success rate. For remove and insert_between action, the success rates increase from 84% to 100%, and 45.6% to 83.0%, but are still limited in hard tasks like rotation: from 6.8% to 18%.
>
> We added more descriptions in Section 4.1 of the main text. Meanwhile, our complementary tests e.g. pointworld serves to simplify the syntax following requirements of the main benchmark, and helps to distinguish reasoning from symbolic manipulation errors.
>
> We thank you for your valuable feedback and comments. We hope our clarifications address the concerns raised.

---

### Official Review · Reviewer_iYfW · 2025-11-03

**Soundness:** 2
**Presentation:** 3
**Contribution:** 3
**Rating:** 4
**Confidence:** 4

**Summary:**

The proposed AtomWorld is a benchmark designed to evaluate spatial reasoning capabilities of LLMs in the domain of crystalline materials. The authors assume that LLMs possess the capability to understand and manipulate 3D atomic structures for crystal design. They designed a series of cognitive tasks in materials discovery and benchmarked several state-of-the-art LLMs, including Gemini 2.5 Pro, GPT-o3, GPT-o4-mini, Deepseek Chat, Llama-3 70B, and Qwen-3. All benchmark tasks uses CIF format as textual input of the LLMs along with a natural language action prompt, requiring certain actions to edit the CIF in order to output a valid modified CIF structure. The benchmark results suggest high success rates on simple operations like adding or changing single atoms, while failures on rotations and atom swapping. The work try to disentangle different failure modes, but the analysis suggests LLMs are tackling the tasks as pattern-matching rather than physics-informed reasoning.

**Strengths:**

1. The work aims at building a comprehensive benchmark that assesses whether LLMs can handle mechanical operations and higher-level cognitive tasks. It is working at building the foundation for future material discovery with LLM capabilities.
2. Multiple LLMs across different model families are benchmarked in this work. And it reported an interesting observation on how all LLMs consistently fail at complex spatial reasoning tasks.

**Weaknesses:**

1. The benchmarking data are oversimplified. This work evaluates LLMs exclusively on simple prototype crystals with at most three elements, which raises questions about whether the findings generalize to multi-component systems or more complex lattice structures.
2. In addition, the benchmark rely on extremely small sample sizes often tens or hundreds. E.g. CIF-Repair was tested on 22 samples, CIF-Gen on 20 samples, and only 10 samples ran DFT in StructProp. Therefore the conclusions drawn from these tests about model capabilities or reasoning quality are not reliable.
3. The StructProp benchmark design is intuitive but fundamentally flawed, as it assumes LLMs understand the property concepts being tested without validating this assumption. Interpretation of the StructProp results can only be regarded as bias from memorized patterns.

**Questions:**

1. All demonstrated examples use relatively simple crystals with at most three elements. Have you benchmarked LLMs on materials with higher compositional complexity or complex crystal systems, e.g. from the Materials Project database?
2. The CIF format contains many redundant information and can produce lengthy token sequences. Have you evaluated the LLMs' performance on token length of the CIFs? How does the token counts distribute for each task? How does performance correlate with the length of the input CIF file? Does the context window limitations affect performance of the LLMs?
3. Specifically when atom coordinates involved, have you tested whether generating a structure where the target atom appears early or late in the CIF sequence lead to different results?
4. For StructProp, have you tested whether models can correctly identify which property is being referenced?

---

> ### Author Response · Authors · 2025-11-23
>
> **Part1:**
>
> Thank you for your feedback and we would like to offer some clarifications on points that could have led to misunderstandings:
>
> **Q1:** The benchmarking data are oversimplified. This work evaluates LLMs exclusively on simple prototype crystals with at most three elements, which raises questions about whether the findings generalize to multi-component systems or more complex lattice structures. **(For Weakness 1 & Question 1)**
>
> **A1:** Thank you for raising this. We would like to clarify that our evaluations include crystals beyond simple prototypes and with more than three elements. The confusion may have arisen from misunderstanding Figure 4 (previously Figure 3) and being confused by our complementary CIF-Gen dataset for the main AtomWorld dataset. We generally cover the CIFs containing 10\~100 atoms, 1\~9 element types, and a wide range of symmetry groups.  In our revised manuscript, Figure 4 now clearly states that it depicts CIF-Gen results. Moreover, we include Figure 2 to explicitly portray the distribution of crystals in AtomMotor-1K. Further, Appendix A.1, covers the number of atoms, number of element types, element distributions, and symmetry-group distributions of our dataset.
>
> **Q2:** In addition, the benchmark rely on extremely small sample sizes often tens or hundreds. E.g. CIF-Repair was tested on 22 samples, CIF-Gen on 20 samples, and only 10 samples ran DFT in StructProp. Therefore the conclusions drawn from these tests about model capabilities or reasoning quality are not reliable. **(For W2)**
>
> **A2:** We would like to clarify the scope of these experiments. The subsets cited (e.g., 10 samples of DFT in StrucProp) were designed as **high-fidelity validation case studies** to corroborate the broader trends, rather than as the primary benchmark metrics. Given the high computational cost of DFT relaxation and manual verification required for CIF-repair, these analyses focused on representative depth rather than breadth.
>
> Importantly, our core AtomWorld benchmark operates on a significantly larger scale. Furthermore, Section 4.3 describes that the sample size is statistically sufficient to yield stable data distribution. We have revised the **introduction and methodology** sections to explicitly distinguish between the large-scale benchmark and these auxiliary deep analyses.
>
> **Q3:** The StructProp benchmark design is intuitive but fundamentally flawed, as it assumes LLMs understand the property concepts being tested without validating this assumption. Interpretation of the StructProp results can only be regarded as bias from memorized patterns. **(For W3 and Q4)**
>
> **A3:** Thank you for this critical observation. We respectfully clarify that StructProp is not based on the assumption that LLMs understand these concepts; rather, it’s designed as a diagnostic tool to test that very hypothesis.
>
> In fact, your point - that results may stem from ‘memorized patterns’- aligns with our findings in Section 5.3. Our analysis shows that LLMs only knows the basic definition and potential influence factors about the property, but lack key and correct reasoning that shows real understanding towards the materials. Therefore, structprop successfully serves its analytical role by distinguishing between the surface-level pattern matching and real physical understanding, showing the limitation you correctly identify.
>
> **Q4:** The CIF format contains many redundant information and can produce lengthy token sequences. Have you evaluated the LLMs' performance on token length of the CIFs? How does the token counts distribute for each task? How does performance correlate with the length of the input CIF file? Does the context window limitations affect performance of the LLMs? **(For Q2)**
>
> **A4:** Thank you for raising this. In our case, we have used a standardized, symmetry-implicit CIF format. Thus, the lines above coordinates are almost identical across the datasets, and the token length is directly related to the number of atoms. So far, we have analysed the sensitivity of our results to the number of atoms in CIFs, which is directly correlated with token length. We have since added an investigation to Appendix B.2.1 of the revised manuscript. In brief, the success rates decrease as the length of the input CIF increases. With the number of atoms going beyond 100 (about 6600 tokens), the success rates go below 20%. As CIF content length increases, LLM performance generally degrades, which sets a ceiling on the maximum usable CIF size well before context window limitations come into play. This indicates that the bottleneck is not token capacity, but rather the model’s ability to maintain long-context reasoning over complex structured data.
>
> Hence in our benchmark we keep the input CIFs as a controlled variable across action classes to ensure fair comparison

---

> ### Author Response · Authors · 2025-11-29
>
> **Part 2:**
>
> **Q5:** Specifically when atom coordinates are involved, have you tested whether generating a structure where the target atom appears early or late in the CIF sequence lead to different results? **(For Q3)**
>
> **A5:** We have added this investigation to Appendix B.2.3 of the revised manuscript.
>
> Table1: The success rate for different action positions.
> | Action position | Deepseek-chat (%) | Qwen3 32B (%) |
> |----------------|------------------|---------------|
> | Index 0, 1     | 90               | 96            |
> | Index -2, -1   | 82               | 80            |
>
> In brief, as shown in Table 1 above, when the action performed on the first few atoms are generally better than those in the last few atoms. Indicating that atom ordering has some impact on performance, but remains limited.
>
> Thank you for the revision. We hope that our responses have satisfactorily addressed all your queries. Should you have further questions or suggestions for enhancing our manuscript, we warmly welcome your input.

---

### Meta-Review · Area_Chair_m5YZ · 2026-01-07

**Summary:**

The paper introduces a benchmark for evaluating spatial reasoning capabilities of LLMs in crystalline materials. Reviewers felt that (1) the benchmark is over simplified with small sample size, and (2) the design may be fundamentally flawed and may introduce bias in the results.  As a result, the benchmark may bring limited benefits.

**Reviewer Concerns:**

The authors provided additional technical details and experiments. but did not expand the benchmark to make it more realistic. nor provided a convincing argument why it was not necessary.

**Reviewer Scores:**

I do not have strong feeling on whether the reviewers will raise their scores.

---

### Decision · Program_Chairs · 2026-01-26

Reject